# PATCH REBIRTH: TOWARD FAST AND TRANSFERABLE MODEL INVERSION OF VISION TRANSFORMERS

## ABSTRACT

Model inversion is a widely adopted technique in data-free learning that reconstructs synthetic inputs from a pretrained model through iterative optimization, without access to original training data. Unfortunately, its application to state-of-the-art Vision Transformers (ViTs) poses a major computational challenge, due to their expensive self-attention mechanisms. To address this, *Sparse Model Inversion* (SMI) was proposed to improve efficiency by pruning and discarding seemingly unimportant patches, which were even claimed to be obstacles to knowledge transfer. However, our empirical findings suggest the opposite: even randomly selected patches can eventually acquire transferable knowledge through continued inversion. This reveals that discarding any prematurely inverted patches is inefficient, as it suppresses the extraction of class-agnostic features essential for knowledge transfer, along with class-specific features. In this paper, we propose *Patch Rebirth Inversion* (PRI), a novel approach that incrementally detaches the most important patches during the inversion process to construct sparse synthetic images, while allowing the remaining patches to continue evolving for future selection. This progressive strategy not only improves efficiency, but also encourages initially less informative patches to gradually accumulate more class-relevant knowledge, a phenomenon we refer to as the *Re-Birth* effect, thereby effectively balancing class-agnostic and class-specific knowledge. Experimental results show that PRI achieves up to $10\times$ faster inversion than standard *Dense Model Inversion* (DMI) and $2\times$ faster than SMI, while consistently outperforming SMI in accuracy and matching the performance of DMI.

## 1 INTRODUCTION

Model inversion (Fredrikson et al., 2015; Mahendran & Vedaldi, 2015; Yin et al., 2020) is a prominent technique in data-free learning, aiming to reconstruct synthetic inputs from a pretrained model via iterative optimization, without using any original inputs. In data-constrained scenarios where the original dataset is unavailable (e.g., due to privacy concerns), the synthesized inputs generated by model inversion can serve as carriers of the model's pretrained knowledge, which can then be transferred into any target model for training. One of the predominant applications is data-free model compression, such as quantization and distillation without using original samples, where training or fine-tuning the compressed model is essential to recover performance degradation. While earlier studies on model inversion have primarily focused on convolutional neural networks (CNNs), the recent rise of Vision Transformers (ViTs) (Dosovitskiy et al., 2021) motivates the development of new approaches that exploit their architectural strengths.

However, a major drawback of model inversion is its high computational overhead, to the extent that generating only a few hundred synthetic images can take several hours even on a high-end GPU[1]. This inefficiency becomes more exacerbated in ViTs, whose complexity substantially increases with the number of tokens, generally exceeding the computational cost of CNNs (He et al., 2016; Krizhevsky et al., 2012). To address this, *Sparse Model Inversion* (SMI) (Hu et al., 2024) was recently introduced, inspired by token pruning (Liang et al., 2022; Rao et al., 2021) that aims to accelerate ViT inference by discarding unimportant tokens. SMI applies a *reversed* strategy by removing less informative

---

[1]On an RTX A6000, it takes about 1 hour to invert just 128 images of 224×224 with DeiT-Base.

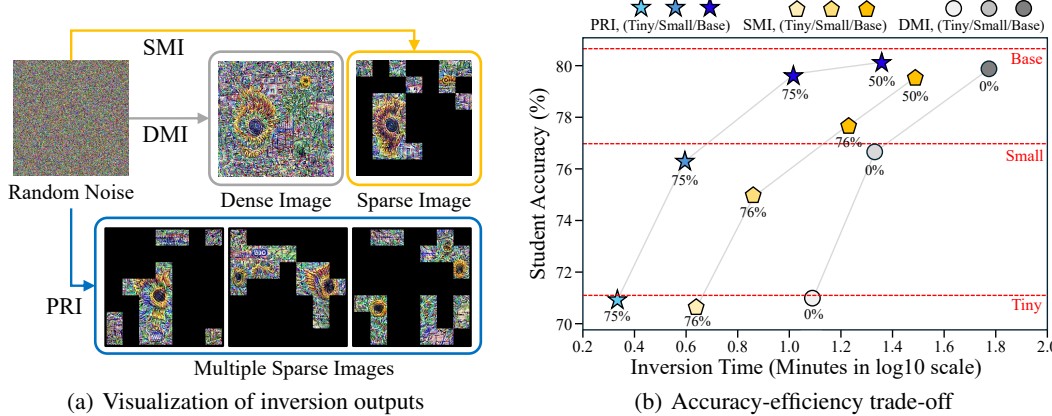

(a) Visualization of inversion outputs        (b) Accuracy-efficiency trade-off

Figure 1: Comparison of DMI (Yin et al., 2020), SMI (Hu et al., 2024), and PRI (ours) on CIFAR-100. (a) Visualization illustrating the differences among the three inversion methods. (b) Student accuracy distilled from the same teacher, DeiT-Tiny/Small/Base; denoted as (T)/(S)/(B), with GPU time (in $\log_{10}$ minutes) measured for inverting 128 samples per batch. Red dashed lines indicate teacher accuracy, and percentages denote image sparsity.

patches during inversion, instead of inference, based on its core hypothesis that these patches are not only redundant but may also hinder effective knowledge transfer.

In this paper, we revisit the core assumption of SMI and argue that retaining all patches during model inversion is indeed more effective, particularly for conveying transferable knowledge in data-free learning. Our first empirical observation is that: unlike token pruning on real images, pruning inverted patches does not strongly depend on patch importance estimated in the initial phase. In our study (see Table 1), even randomly selected patches achieved comparable performance to those selected based on importance by the end of the inversion process. This leads to our key insight: *regardless of their initial importance, any selected patches can ultimately embed highly transferable knowledge through iterative inversion.* From this perspective, SMI's strategy of discarding unimportant patches is not only ineffective for knowledge transfer but ironically also inefficient in terms of inversion time. Once pruned, patches are permanently excluded from synthesis, regardless of any useful knowledge they may have acquired. This further causes the inversion region to gradually shrink, thereby limiting the diversity of synthesized features. To be revealed in our empirical analysis, this behavior leads to overfitting to *class-specific* features, while suppressing *class-agnostic* information, which is crucial for generalizable knowledge transfer in data-free settings.

Based on our findings above, we propose a more efficient yet more effective model inversion method for data-free knowledge transfer, called *Patch Rebirth Inversion* (PRI). Rather than generating a single sparse image through gradual patch pruning, PRI makes full use of the inverted knowledge throughout the inversion process by producing a sequence of sparse images (see Figure 1(a)). Each sparse image is constructed by isolating the most important patches at a specific point of the inversion process. Thus, these sparse images are not generated all at once, but rather progressively separated from the full image over the iterations. Interestingly, we discover that isolating important patches encourages the remaining ones to start synthesizing more meaningful features, a phenomenon we refer to as the *Re-Birth* effect. Some of these *reborn* features eventually become informative enough to form another sparse image at subsequent iterations. This progressive mechanism not only allows the generation of multiple sparse images but also increases the diversity of knowledge embedded in the synthesized images. As a result, under the same computational budget, PRI accelerates the inversion process by enabling the production of more synthetic samples. Furthermore, since each image is extracted at a different point along the inversion trajectory, they jointly capture both class-specific and class-agnostic features, leading to improved transferability.

As summarized in Figure 1(b), PRI consistently lies on the *Pareto-optimal curve* of the accuracy-efficiency trade-off, clearly achieving the most favorable balance among all compared methods. In our detailed experimental results (see Table 2), PRI achieves up to $10\times$ faster inversion than *Dense*

*Model Inversion* (DMI) (Yin et al., 2020), a standard method without any sparsification, and up to 2× faster than SMI, while consistently delivering higher accuracy than SMI and maintaining performance close to DMI despite the substantial speedup. We attribute this superiority to PRI's ability to effectively embed both class-agnostic and class-specific knowledge into the inverted patches, as further supported by our in-depth empirical analysis.

## 2 RELATED WORK

Model inversion has long been studied across a range of contexts, from privacy attacks (Fredrikson et al., 2015; He et al., 2019; Wang et al., 2015; Yang et al., 2019) to the analysis of deep feature representations (Mahendran & Vedaldi, 2015; 2016), commonly aiming to understand and exploit various pretrained models. More recently, it has become a central component in data-free learning, a popular technique that extracts synthetic inputs from a pretrained model, without accessing original training data. Earlier works focus on convolutional neural networks (CNNs), applying model inversion to generate synthetic data for data-free quantization (Cai et al., 2020; Choi et al., 2021; Nagel et al., 2019; Xu et al., 2020; Zhang et al., 2021; Zhong et al., 2022) and knowledge distillation (Binici et al., 2022; Chen et al., 2019; Fang et al., 2019; 2021; Lopes et al., 2017; Shin & Choi, 2024; Yin et al., 2020). These approaches typically utilize convolutional features and batch normalization statistics to enhance the performance.

**Model Inversion in ViTs.** With the popularity of Vision Transformers (ViTs), which lack batch normalization and exhibit unique architectural properties, alternative model inversion strategies have been proposed to exploit their patch-wise and self-attention mechanism. Among various attempts (Choi et al., 2025; Li et al., 2022; 2024; Ramachandran et al., 2024) to adapt model inversion to ViTs, PSAQ-ViT (Li et al., 2022) first introduced a patch similarity-aware strategy for data-free quantization by leveraging self-attention scores to identify redundant tokens and guide quantization accordingly. MimiQ (Choi et al., 2025) further explored data-free quantization for ViTs, observing that alignment of attention maps between teacher and student models significantly enhances recovery of the performance in the quantized model. Despite these efforts, they all adopt dense inversion strategies that optimize every patch simultaneously during the entire process and therefore suffer from substantial computational cost, due to the high computational complexity with respect to the number of patches.

**Sparse Model Inversion.** To address this inefficiency, sparse model inversion (SMI) (Hu et al., 2024) was recently introduced, inspired by token pruning strategies (Kim et al., 2022; Liang et al., 2022; Rao et al., 2021; Wang et al., 2021). Instead of updating all patches, SMI selectively inverts only a subset of important patches to reduce computational overhead. This patch selection process assumes that tokens with low attention contribute little to knowledge transfer, and should be discarded as early as possible. While this may offer some efficiency gains, it overlooks a key opportunity: previously inverted patches, even if initially deemed unimportant, may still carry transferable features. According to our empirical study, discarding these patches limits the representational diversity of synthetic images. In contrast, allowing all patches to remain involved throughout the inversion, regardless of their initial importance, enables the gradual emergence of both class-specific and class-agnostic features that are essential for effective knowledge transfer.

## 3 PRELIMINARIES

This section provides a formal definition of model inversion in the context of ViTs, along with a description of the attention-based token selection mechanism adopted in both SMI (Hu et al., 2024) and our proposed method.

**Formulation.** Given a pretrained classification model $f$, model inversion aims to synthesize input images that reflect the knowledge learned by the model, without access to the original training data. Formally, for a target label $y \in \{1, \ldots, c\}$ and a randomly initialized image $\hat{\mathbf{X}} \in \mathbb{R}^{H \times W \times C}$ (where $H$, $W$, and $C$ denote height, width, and the number of channels, respectively), $\hat{\mathbf{X}}$ is iteratively updated by minimizing the following inversion loss:

$$\mathcal{L}_{\text{inv}}(\hat{\mathbf{X}}, y; f) = \mathcal{L}_{\text{cls}}(f(\hat{\mathbf{X}}), y) + \lambda \mathcal{L}_{\text{reg}}(\hat{\mathbf{X}}), \tag{1}$$

where $\mathcal{L}_{\text{cls}}$ is a classification loss that encourages the image to be predicted as class $y$, and $\mathcal{L}_{\text{reg}}$ is a regularization term to enhance visual plausibility. As adopted by many existing works (Braun et al., 2024; Hatamizadeh et al., 2022; Hu et al., 2024; Yin et al., 2020), we use cross-entropy for $\mathcal{L}_{\text{cls}}$ and total variation (TV) regularization for $\mathcal{L}_{\text{reg}}$.

**ViT Inversion.** In the context of ViTs, an image $\hat{\mathbf{X}}$ needs to be divided and flattened into a sequence of $N$ disjoint patches, denoted by $\{\mathbf{x}_j\}_{j=1}^N$, where each patch $\mathbf{x}_j$ is of size $P \times P$ (i.e., $\mathbf{x}_j \in \mathbb{R}^{P \times P \times C}$), and consequently $N = \frac{H \times W}{P^2}$. These patches are then linearly projected into patch embeddings, augmenting with positional encodings (to capture their spatial relationships) and a `[CLS]` token (a special token for class prediction). This augmented sequence of patches are passed through $L$ transformer encoder layers. Each encoder layer consists of multi-head self-attention (MHSA) and feed-forward networks (FFNs), where MHSA computes scaled dot-product attention, expressed as:

$$\text{Attention}(\mathbf{Q}, \mathbf{K}, \mathbf{V}) = \text{Softmax}\left(\frac{\mathbf{Q}\mathbf{K}^T}{\sqrt{d}}\right)\mathbf{V}, \tag{2}$$

where queries ($\mathbf{Q}$), keys ($\mathbf{K}$), and values ($\mathbf{V}$) are linear projections of input embeddings, and $d$ is the embedding dimension. The computational complexities of MHSA and FFN are given by $\mathcal{O}(SA) = 4Nd^2 + 2N^2d$ and $\mathcal{O}(FFN) = 8Nd^2$ (Chen et al., 2023b), respectively. Since the inversion process involves repeated forward and backward passes to minimize Eq. (1), the total computational cost of ViT inversion over $T$ iterations and $I$ images across $L$ layers is represented as:

$$\mathcal{C}_{\text{DMI}}^{\text{SA}} = L \cdot (4Nd^2 + 2N^2d) \cdot I \cdot T,$$
$$\mathcal{C}_{\text{DMI}}^{\text{FFN}} = L \cdot 8Nd^2 \cdot I \cdot T.$$

where $\mathcal{C}_{\text{DMI}}^{\text{SA}}$ and $\mathcal{C}_{\text{DMI}}^{\text{FFN}}$ represent the total costs of MHSA and FFN layers, respectively. Therefore, minimizing the number of patches is crucial for improving the overall efficiency of ViT inversion.

**Patch Selection via Attention Scores.** To improve the efficiency of ViT-based methods, token (or patch) selection has become a common strategy, based on token importance. A standard practice for estimating importance is to leverage attention scores, which are derived from the matrix $\frac{\mathbf{Q}\mathbf{K}^T}{\sqrt{d}}$ in Eq. (2), and to take the average over the scores from the `[CLS]` token to all other tokens, considering how much each patch contributes to the model's prediction. Prior works such as SMI leverage these importance scores to discard less important tokens, thereby reducing computational overhead. Particularly in SMI, inverted patches that are deemed unimportant are removed early from the optimization process, with the goal of accelerating inversion while preserving essential information. In contrast, our PRI method also employs attention scores to identify important patches but does not discard unimportant ones; instead, it retains them for subsequent inversion iterations.

## 4 METHODOLOGY

In this section, we present our proposed method, Patch Rebirth Inversion (PRI), designed to improve both the efficiency and effectiveness of ViT-based model inversion.

### 4.1 REVISITING PATCH PRUNING IN MODEL INVERSION

We begin by revisiting the fundamental assumption underlying the Sparse Model Inversion (SMI) approach (Hu et al., 2024), particularly the claim that early removal of low-importance patches not only accelerates inversion process but also benefits the effectiveness of knowledge transfer. Through our empirical studies, we uncover two key observations that challenge this assumption: (1) the diminishing impact of patch selection as inversion progresses, and (2) the late emergence of meaningful features from initially unimportant patches, a phenomenon we call the *Re-Birth* effect.

**Limited Impact of Selection Criterion.** Our first investigation is about how strongly the choice of patch selection criterion affects inversion effectiveness. In addition to high-attention selection, we evaluate several seemingly ineffective strategies, namely low-attention, random, and fixed-region (top) patch selection, where the selected patches remain unpruned until the end of the inversion

Table 1: Knowledge distillation performance under different patch selection strategies in SMI: high-attention, low-attention, random, and fixed-region (top), where DeiT-Base is fine-tuned on 32 inverted CIFAR-10 images with 76% sparsity for 120 epochs.

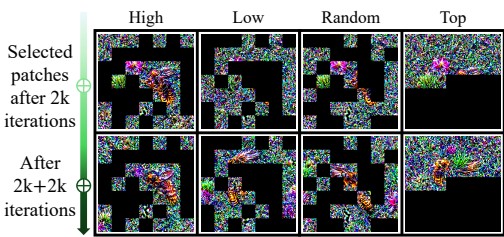

Figure 2: Illustration of the Re-Birth effect: inverted images produced by four different patch selection strategies, shown after 2k iterations (top) and 4k iterations (bottom).

| Dataset: **CIFAR-10** (Teacher: DeiT-Base, Acc: 95.4) | | | | |
|---|---|---|---|---|
| Patch Selection Sparsity | High 76% | Low 76% | Random 76% | Top 76% |
| Model │ DeiT-Base | 92.52 | 92.75 | 92.74 | 92.74 |

process. Unlike our expectation of noticeable differences in downstream performance, our empirical results in Table 1 indicate that, given the same number of inverted images, all selection strategies yield nearly identical performance. This unexpected outcome reveals that the impact of patch selection becomes saturated as the inversion process continues, to the extent that even randomly selected patches can lead to competitive performance.

**Re-Birth Effect.** The counter-intuitive result above naturally raises the following question: *how can initially less important patches achieve performance nearly identical to those selected based on high importance?* By thoroughly visualizing intermediate inverted images, we discover an interesting phenomenon that initially uninformative patches undergo significant transformation when inversion continues beyond the early selection phase. As shown in Figure 2, the high-attention case exhibits little change over time (the bee was already visible after 2k iterations). In contrast, the fixed-region (top) selection approach, which initially lacked recognizable content, regenerates clear bee semantics in the remaining patches. Even low-attention and random selection approaches recover semantic details across disorganized patches. We term this phenomenon the *Re-Birth Effect*, where prolonged inversion allows previously low-importance patches to gradually accumulate meaningful class-relevant features.

These empirical findings demonstrate that the main strategy of SMI, stopping inversion early and discarding unimportant patches, must be revisited in terms of both efficiency and effectiveness. By prematurely stopping inversion for certain patches, SMI prunes valuable semantic knowledge that these patches could accumulate over additional iterations. Furthermore, this restrictive pruning biases the synthesized images towards predominantly class-specific features, while neglecting class-agnostic features essential for robust knowledge transfer, as revealed by our empirical study.

### 4.2 PATCH REBIRTH INVERSION

Motivated by the discoveries above, we propose a fundamentally different approach, PRI, which *enables patch rebirth* throughout the inversion process, where even initially unimportant patches are given the opportunity to be *reborn* through continued inversion iterations. To this end, our method alternates between two operations during the inversion process: (1) detachment of most important patches to be stored as independent sparse images, and (2) continued inversion on the remaining patches, allowing them to evolve and eventually qualify for detachment in future iterations.

**Detachment of Important Patches.** As opposed to SMI (Hu et al., 2024), which discards unimportant patches from their process, PRI detaches the most important patches at specific points of the inversion process, thereby stopping their optimization in subsequent iterations. These detached patches are then stored separately to form an independent sparse image as one of the final outputs of the inversion process. More specifically, as illustrated in Figure 3, consider the first detachment point $t_1$ and its corresponding sequence of inverted patches, $\{\mathbf{x}_j^{(t_1)}\}_{j=1}^N$. At this point, we compute patch-wise importance scores using the attention-based metric. We then identify the top-$K$ patches with the highest importance, where $K < N$ is a parameter that determines the target sparsity and is set according to our detachment scheduling policy (detailed below). These top-$K$ patches are detached from the full patch set to form an independent sparse synthetic image, denoted as $\hat{\mathbf{X}}_{t_1}$. The same procedure is applied at subsequent detachment points $t_2, t_3, \ldots$, yielding non-overlapping

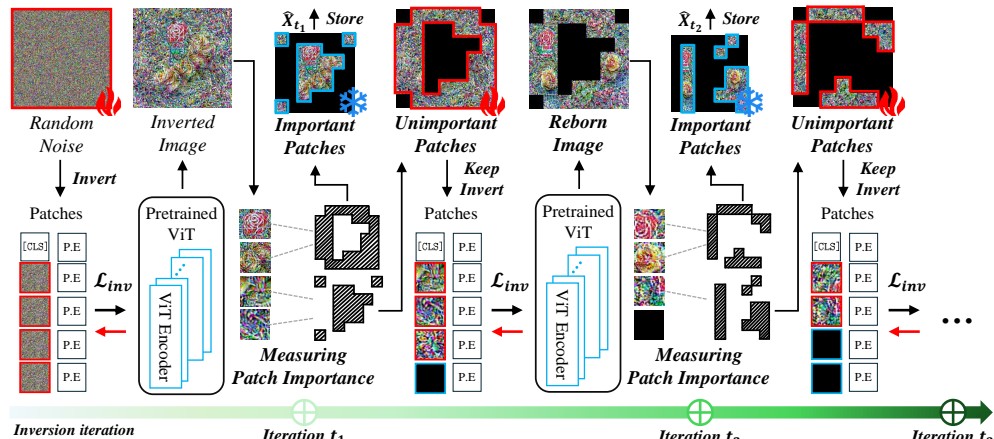

Figure 3: Overview of patch rebirth inversion. At each iteration $t_i$, we store blue framed important patches and mask them out (black) while the remaining red framed patches continue inversion, progressively embedding class-specific features. All stored sparse view compose the final synthesized dataset, which is used for data-free downstream tasks.

sparse images $\hat{\mathbf{X}}_{t_2}, \hat{\mathbf{X}}_{t_3}, \ldots$. Notably, since these images are synthesized at different stages of the inversion process, they are expected to capture varying levels of class-agnostic features (e.g., shared background elements) and class-specific features (e.g., distinct object regions). Nonetheless, as each image consists of the most important patches at its corresponding detachment point, all are expected to contain meaningful knowledge for downstream tasks.

**Inversion of Remaining Patches.** At each detachment point $t_k$, after the top-$K$ important patches are removed, the remaining patches, which have been deemed less important until the point, continue to be optimized in the subsequent iterations. For instance, in Figure 3, after $t_1$, the remaining $N - K$ patches undergo continued inversion until $t_2$, at which point another top-$K$ subset is detached to form $\hat{\mathbf{X}}_{t_2}$. This process will be repeated from $t_2$ to $t_3$, where another remaining set of $N - 2K$ patches will continue to get forward pass for further inversion. Note that these remaining patches after $t_2$ are likely to be more class-specific at $t_3$, even if they start with less informative features than those of selected at $t_2$. As a result, this progressive inversion strategy substantially diversifies and enriches all the generated output images, not only within each instance (as examined by Figure 4) but also across different images, making them highly effective for downstream knowledge transfer.

**Sparsity Control.** To control the number of patch detachments, we define a division factor $v$, which determines into how many partitions a full-size image will be split. Thus, $v$ is not a hyperparameter, but a control parameter adjusting the target sparsity (i.e., $sparsity = 1 - \frac{1}{v}$). For instance, for PRI to achieve 75% sparsity, we need to set $v = 4$. Specifically, given the total number of inversion iterations $T$, we define the detachment points as:

$$t_k = k \cdot \left\lfloor \frac{T}{v} \right\rfloor, \quad \text{for } k \in \{1, 2, \ldots, v\}.$$

According to this policy, $v$ also specifies how many sparse images will be generated during progressive inversion, where each sparse image has the same sparsity level equally containing $K$ patches (as mentioned above), leading to $K = \lfloor \frac{N}{v} \rfloor$ except for the final point $t_v$ that will return all $N - K(v-1)$ remaining patches. Over $T$ iterations, $v$ sparse images are sequentially generated, each representing the most informative content synthesized at different inversion points.

## 4.3 Theoretical Analysis on Inversion Cost

We finally provide a theoretical analysis that supports the computational efficiency of PRI, by comparing its cost against those of DMI and SMI. To this end, we adopt the standard complexity formulations of the MHSA and FFN layers in ViTs, $\mathcal{O}(SA) = 4Nd^2 + 2N^2d$ and $\mathcal{O}(FFN) = 8Nd^2$, respectively (Chen et al., 2023b), and express the total complexity of each inversion method over

$T$ iterations and $I$ images across $L$ layers. For simplicity, we consider an idealized version of SMI, denoted as SMI$^*$, which assumes that inversion starts directly with a reduced set of patches, without gradual pruning over iterations. Even under this optimistic assumption, the following theorem shows that PRI incurs the lowest computational cost in both MHSA and FFN layers.

**Theorem 1.** *(Inversion cost ordering).* *Given $\mathcal{O}(SA) = 4Nd^2 + 2N^2d$ and $\mathcal{O}(FFN) = 8Nd^2$ in ViTs, where $N$ is the number of all patches and $d$ is the embedding dimension, for the overall forward–backward costs under three inversion methods, denoted by $\mathcal{C}^{\mathrm{SA}}$ and $\mathcal{C}^{\mathrm{FFN}}$, it holds that:*

(a) **SA modules.** $\mathcal{C}^{\mathrm{SA}}_{\mathrm{PRI}} < \mathcal{C}^{\mathrm{SA}}_{\mathrm{SMI}^*} < \mathcal{C}^{\mathrm{SA}}_{\mathrm{DMI}}$ *whenever $\frac{N}{d} < 3$, i.e., PRI achieves the lowest cost under the practical condition $N < 3d$ satisfied by standard ViT architectures.*

(b) **FFN modules.** $\mathcal{C}^{\mathrm{FFN}}_{\mathrm{PRI}} < \mathcal{C}^{\mathrm{FFN}}_{\mathrm{SMI}^*} < \mathcal{C}^{\mathrm{FFN}}_{\mathrm{DMI}}$, *where the relative gain of PRI over SMI$^*$ increases with the division factor $v$ and asymptotically approaches $2\times$.*

*Proof.* See Appendix for detailed derivations. ∎

## 5 EXPERIMENTS

In this section, we empirically validate the performance of our PRI method by exploring the following three questions: (1) how much PRI improves inversion efficiency, compared to standard dense inversion (DMI) as well as its faster state-of-the-art variant, SMI (Hu et al., 2024); (2) whether the synthetic images inverted by PRI lead to better knowledge transfer in two prominent data-free learning tasks, namely quantization and distillation; and finally (3) how and why PRI extracts more transferable knowledge through its progressive inversion process.

**Experimental Setup.** Adopting the existing setup (Hu et al., 2024), we use DeiT (Touvron et al., 2021) models with a patch size of 16 from the `timm` library (Wightman, 2019) as the backbone models for inversion. All images are inverted using Adam for 4,000 iterations with a learning rate of 0.25. The hyperparameter $\lambda$ for the inversion loss in Eq. (1) is set to $10^{-4}$, following standard practice (Yin et al., 2020). For the default sparsity of inverted images, we also follow the original SMI setting (i.e., 76%) by applying pruning at iterations 50, 100, 200, and 300 with the same ratio of 0.3. To match this 76% sparsity in PRI, we set $v = 4$, which yields 75% sparsity according to our detachment policy. All experiments were conducted on a single NVIDIA RTX A6000 GPU. Full details are provided in the Appendix.

### 5.1 INVERSION EFFICIENCY

In Table 2, we report the inversion throughput (i.e., the number of iterations per second), computational cost (FLOPs), and GPU memory consumption of different inversion methods using DeiT architectures when synthesizing 128 images per batch. As theoretically proved in Theorem 1, PRI achieves up to $2\times$ faster inversion than SMI and $10\times$ faster inversion than DMI as the division factor $v$ increases. PRI also reduces FLOPs by up to 50% and GPU memory usage by up to 60% compared to SMI. Importantly, the efficiency gains of PRI become more notable at higher sparsity levels, yielding increasingly larger margins over SMI. Overall, these empirical results demonstrate that PRI is significantly more efficient than both DMI and the state-of-the-art SMI method.

### 5.2 EFFECTIVENESS IN DATA-FREE KNOWLEDGE TRANSFER

Given the superior efficiency of PRI shown in Table 2, we now evaluate how effectively the inverted images convey pretrained knowledge in two prominent data-free knowledge transfer tasks, quantization and knowledge distillation.

**Quantization.** Table 3(a) presents the resulting accuracy of *quantization-aware training* (QAT), where 10k inverted images from DeiT-Base are used to fine-tune quantized models for 100 epochs with a learning rate of 0.001. Specifically, we adopt *learned step size quantization* (LSQ) (Esser et al., 2020) for fine-tuning, using only inverted images without access to the original training data. Despite limited room for improving over the original model accuracy, PRI even outperforms DMI at 50% sparsity and thus achieving faster inversion, and shows only a minor accuracy drop at 86%

Table 2: Inversion efficiency on DeiT-Base across various sparsity levels. Throughput is the inversion speed, measuring inversion iterations per second. The changes in red and blue refer to the comparison with each sparsity level of SMI. $v$ is division factor. More results are included in the Appendix.

| Method | DMI | SMI | | | PRI | | |
|---|---|---|---|---|---|---|---|
| Sparsity | 0% | 50% | 76% | 86% | 50% ($v=2$) | 75% ($v=4$) | 86% ($v=7$) |
| Throughput (its/s) ↑ | 1.10 | 2.20 | 3.92 | 5.58 | 2.88 (+30.9%) | 6.40 (+63.3%) | **11.81** (+111.6%) |
| FLOPs (T) ↓ | 13.43 | 6.74 | 3.45 | 2.13 | 5.02 (-25.5%) | 2.09 (-39.4%) | **1.07** (-49.8%) |
| GPU Memory (GB) ↓ | 23.42 | 10.77 | 6.26 | 4.61 | 8.99 (-16.5%) | 4.28 (-31.6%) | **2.68** (-41.9%) |

Table 3: Downstream task results on data-free quantization and knowledge distillation using DMI, SMI, and PRI across various sparsity levels. (a) Quantization results on ImageNet-1k, where W4/A8 refers to the bit precision for weight and activation quantization, respectively. (b) Knowledge distillation results on CIFAR-100. The changes in red refer to the comparison with each sparsity level of SMI. $v$ is division factor. More results are included in the Appendix.

| (a) Quantization Results – **ImageNet-1k** (Original: DeiT-Base (32 bits), Acc: 81.7%) | | | | | | | |
|---|---|---|---|---|---|---|---|
| Method | DMI | SMI | | | PRI | | |
| Sparsity | 0% | 50% | 76% | 86% | 50% ($v=2$) | 75% ($v=4$) | 86% ($v=7$) |
| Quantized W4/A8 | 80.19 | 80.29 | 79.77 | 79.20 | **80.36** (+0.07) | 80.13 (+0.46) | 80.07 (+0.87) |
| Accuracy (%) W8/A8 | 80.73 | 80.77 | 80.33 | 79.85 | **80.78** (+0.01) | 80.70 (+0.37) | 80.57 (+0.72) |

| (b) Knowledge Distillation Results – **CIFAR-100** (Teacher: DeiT-Base, Acc: 80.6%) | | | | | | | |
|---|---|---|---|---|---|---|---|
| Method | DMI | SMI | | | PRI | | |
| Sparsity | 0% | 50% | 76% | 86% | 50% ($v=2$) | 75% ($v=4$) | 86% ($v=7$) |
| Student DeiT-Tiny | 54.90 | 48.34 | 24.31 | 3.55 | 54.57 (+6.23) | 43.32 (+19.01) | 21.27 (+17.72) |
| Accuracy (%) DeiT-Small | 67.62 | 62.55 | 45.05 | 11.25 | **67.70** (+5.15) | 62.93 (+17.87) | 45.59 (+34.34) |
| DeiT-Base | 79.76 | 79.41 | 77.55 | 70.22 | **79.98** (+0.57) | 79.57 (+1.98) | 78.46 (+8.24) |

sparsity, where PRI achieves a $10\times$ speedup over DMI in Table 2. Compared to SMI, PRI consistently maintains larger accuracy margins, especially as the sparsity level increases.

**Distillation.** Table 3(b) presents the results of knowledge distillation, where 128 images per batch are inverted to construct a synthetic training set for student models. Each batch is used only once, and no access to original training data is allowed. The teacher model is DeiT-Base pretrained on ImageNet (Deng et al., 2009) yet fine-tuned on CIFAR-100 (Krizhevsky & Hinton, 2009), while the student models are DeiT models pretrained only on ImageNet. Aligning with the QAT results in Table 3(a), PRI even outperforms DMI at 50% sparsity and achieves comparable performance at higher sparsity levels when distilling into the DeiT-Base student. In contrast, when distilling into smaller student models, such as DeiT-Tiny, using highly sparse inverted images (i.e., 75% and 86% sparsity) becomes more challenging, as DMI clearly outperforms both SMI and PRI at 86% sparsity. Nevertheless, PRI still manages to achieve performance close to DMI at 50% sparsity even with the DeiT-Tiny student, and consistently surpasses SMI by a large margin in all cases. SMI, in particular, abruptly fails to transfer knowledge to smaller architectures at higher sparsity levels, showing a sharp degradation in performance.

In summary, PRI enables highly effective data-free knowledge transfer, consistently outperforming SMI and matching or exceeding DMI across both quantization and distillation tasks, even under high sparsity and thus faster inversion.

## 5.3 TRANSFERABILITY ANALYSIS OF PRI

To further understand how and why PRI extracts more transferable knowledge through its progressive inversion process, we investigate its ability to preserve class-agnostic information and support generalized knowledge transfer beyond class-specific reconstruction.

**One-Class Distillation.** We first examine whether PRI can effectively transfer class-agnostic knowledge. To this end, we design an extreme scenario, called *one-class distillation*, where knowledge distillation is performed using inverted images, all corresponding to a single class (airplane in CIFAR-10). Somewhat surprisingly, as shown in Figure 4, the student trained on only "airplane"

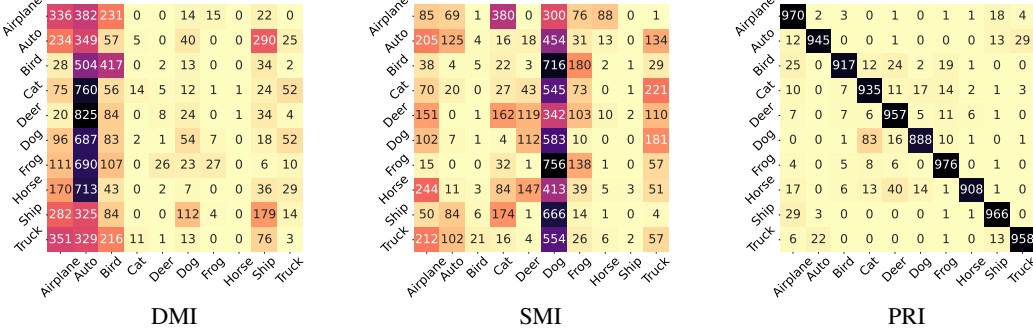

DMI                                    SMI                                    PRI

Figure 4: Confusion matrices of student models trained exclusively on inverted images from a single class, "airplane", in CIFAR-10, using different inversion methods. The architecture of both teacher and student is DeiT-Base. While students trained with DMI and SMI fail to generalize beyond the target class, the student trained with PRI-inverted images exhibits broad generalization across all classes.

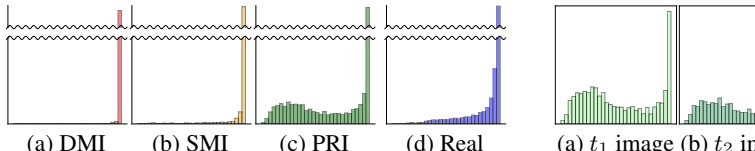

(a) DMI        (b) SMI        (c) PRI        (d) Real

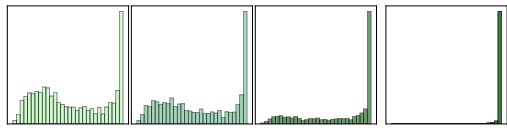

(a) $t_1$ image (b) $t_2$ image (c) $t_3$ image (d) $t_4$ image

Figure 5: Distribution of pretrained teacher model's confidence for 10k synthetic and real images on CIFAR-10. X-axis is confidence and Y-axis is frequency.

Figure 6: Distribution of pretrained teacher model's confidence for 2.5k sparse images of PRI across detachment points $t_1$ to $t_4$. X-axis is confidence and Y-axis is frequency.

images inverted via PRI achieves reasonably strong performance across all 10 classes. In contrast, as intuitively expected, both DMI and SMI fail to capture generalized knowledge across classes, probably due to the fact that their inverted images predominantly encode class-specific features. These results suggest that PRI can extract knowledge that is not only class-agnostic but also transferable even through single-class inversion.

**Confidence Analysis.** Next, generating 10k inverted images evenly across all 10 classes in CIFAR-10, we also report the distributions of the maximum class probabilities (i.e., confidences) predicted by the teacher model. As shown in Figure 5, only PRI exhibits a wide and smooth confidence distribution, showing a level of smoothness comparable to that observed in real images. In contrast, both DMI and SMI yield overly confident predictions, with most values concentrated near 1.0. This also confirms that their inverted images predominantly encode class-specific features while overlooking class-agnostic information.

**Progressive Shift from General to Specific.** Finally, we analyze how PRI gradually transitions from capturing general, class-agnostic knowledge to more specific, class-dependent features during its progressive inversion process. With a division factor $v = 4$, we extract 2.5k inverted images at each of four detachment points, $t_1, ..., t_4$, and present the confidence distributions of the images corresponding to each detachment point. As shown in Figure 6, the confidence distributions gradually shift from the smooth and broad at $t_1$ to the sharp and peaked at $t_4$, reflecting accumulation of class-specific features. This progressive transition enables PRI to retain a broader spectrum of features throughout inversion, in contrast to SMI, which focuses only on class-dependent information.

## 6 CONCLUSION

Motivated by our empirical finding that patches initially considered unimportant can become informative through continued inversion, we proposed Patch Rebirth Inversion (PRI), a method that efficiently synthesizes multiple sparse images capturing both class-agnostic and class-specific features. Extensive experiments demonstrated that PRI significantly accelerates inversion while consistently achieving strong performance across data-free quantization and knowledge distillation tasks.

## REPRODUCIBILITY STATEMENT

We have taken several measures to ensure the reproducibility of our results. All implementation details, including model architectures, training procedures, and hyperparameter configurations, are described in Section 5 of the main text and Appendix D. We provide a complete description of the overall pipeline in Figure 3. To further facilitate reproducibility, we will provide an anonymized GitHub link: `https://anonymous.4open.science/r/PRI-4C56`. Additionally, all baseline methods are implemented using publicly available codes and hyperparameters are carefully tuned following the guidelines in their original papers. Finally, proof of the theoretical claim is included in Appendix F.

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

TECHNICAL APPENDICES

PATCH REBIRTH: TOWARD FAST AND TRANSFERABLE MODEL INVERSION OF VISION TRANSFORMERS

In this Appendix, we provide additional materials supporting our main paper. We begin by reviewing additional related work and offering a detailed explanation of the inversion loss introduced in Eq. (1). Next, we provide the pseudocode for our proposed patch rebirth inversion algorithm. Further experimental details, including experimental setups and hyperparameters, are elaborated. We present supplementary experimental results, encompassing analyses of image properties at each detachment point, t-SNE visualizations of inverted features, and extended quantitative evaluations. The proof for Theorem 1 is presented. Finally, we describe visualization methodologies, and provide additional visualizations demonstrating the re-birth effect.

## A  MORE RELATED WORKS

**Class-Specific and Class-Agnostic Features.**  Recent works in representation learning have focused on disentangling class-agnostic and class-specific features to improve generalization and transferability across various tasks (Stergiou et al., 2020; Zhang et al., 2023). Class-specific features are typically aligned with discriminative information tightly coupled with a particular class, while class-agnostic features capture generic patterns such as texture, shape, and structure that are useful across classes. A class-specific attention mechanism was proposed to highlight discriminative temporal features and improve time-series classification performance across multiple classes (Hao et al., 2022). To mitigate biased classification in few-shot segmentation, an adaptive prototype alignment method was introduced, combining class-specific and class-agnostic prototypes to enhance feature comparisons and generalization (Chen et al., 2023a). While these works have explored class-aware and class-invariant representations in supervised settings, our work investigates how such distinctions naturally emerge in the process of model inversion. Unlike prior work that explicitly disentangles these two types via architectural designs or supervision, we show that different inversion sequences can implicitly control the balance of these features, which is especially important in data-free scenarios.

## B  DETAILED EXPLANATION OF INVERSION LOSS

In this section, we provide a detailed description of each loss component used for model inversion in Eq. (1), specifically the classification loss $\mathcal{L}_{\text{inv}}$ and the regularization loss $\mathcal{L}_{\text{reg}}$.

**Classification Loss.**  Following prior inversion methods (Hu et al., 2024; Yin et al., 2020), we adopt the standard cross-entropy loss as our classification loss, defined as:

$$\mathcal{L}_{\text{cls}}(f(\hat{\mathbf{X}}), y) = -\sum_{i=1}^{c} \mathbb{I}[i = y] \cdot \log \left( \frac{\exp(f(\hat{\mathbf{X}}))}{\sum_{j=1}^{c} \exp(f_j(\hat{\mathbf{X}}))} \right),$$

where $f(\hat{\mathbf{X}}) \in \mathbb{R}^c$ represents the output logits from the pretrained classifier $f$, $c$ is the total number of classes, and $y$ denotes the target class for the inverted image. This loss encourages the synthesized image $\hat{\mathbf{X}}$ to be confidently classified as the target class $y$.

**Regularization Loss.**  For visual plausibility, we adopt total variation (TV) regularization, commonly utilized to encourage smoothness (Hatamizadeh et al., 2022). TV regularization is formally expressed as:

$$\mathcal{L}_{\text{reg}}(\hat{\mathbf{X}}) = \sum_{i=2}^{H} \sum_{j=2}^{W} \left( \left\| \hat{\mathbf{X}}_{i,j} - \hat{\mathbf{X}}_{i-1,j} \right\|_2 + \left\| \hat{\mathbf{X}}_{i,j} - \hat{\mathbf{X}}_{i,j-1} \right\|_2 \right.$$

$$\left. + \left\| \hat{\mathbf{X}}_{i,j} - \hat{\mathbf{X}}_{i-1,j-1} \right\|_2 \right) + \sum_{i=2}^{H} \sum_{j=1}^{W-1} \left\| \hat{\mathbf{X}}_{i,j} - \hat{\mathbf{X}}_{i-1,j+1} \right\|_2,$$

where $\hat{\mathbf{X}}_{i,j}$ denotes the pixel value at spatial coordinates $(i, j)$ of the inverted image. By penalizing large intensity changes between adjacent pixels, this loss term significantly improves the naturalness and continuity of the generated images.

Combining these two terms with a balancing hyperparameter $\lambda$, we obtain the overall inversion loss used throughout our experiments:

$$\mathcal{L}_{\text{inv}}(\hat{\mathbf{X}}, y; f) = \mathcal{L}_{\text{cls}}(f(\hat{\mathbf{X}}), y) + \lambda \mathcal{L}_{\text{reg}}(\hat{\mathbf{X}}).$$

As mentioned in the main paper, we set $\lambda = 10^{-4}$, following standard practice (Yin et al., 2020).

## C PSEUDOCODE OF PATCH REBIRTH INVERSION

To clearly present the implementation details of our approach, we provide the pseudocode of patch rebirth inversion in Algorithm 1. The algorithm describes how PRI progressively stores important patches as sparse images at each detachment point while continuing to invert the remaining unimportant patches. The formulation follows the notation introduced in the preliminaries section and omits auxiliary regularization or architectural specifics for clarity.

---

**Algorithm 1** Patch Rebirth Inversion (PRI)

---

**Input:** Pretrained model $f$, total iterations $T$, division factor $v$, random noise $\hat{\mathbf{X}}_{t_0}$, target label $y$
**Output:** Set of sparse synthetic images $\mathcal{S}$
1: Initialize: $\mathcal{S} \leftarrow \emptyset$, $\hat{\mathbf{X}} \leftarrow \hat{\mathbf{X}}_{t_0}$, $K \leftarrow \lfloor \frac{N}{v} \rfloor$, active patch set $\mathcal{P} \leftarrow \{\mathbf{x}_1, \ldots, \mathbf{x}_N\}$
2: **for** $t \leftarrow 1$ **to** $T$ **do**
3:     **if** $t = k \cdot \lfloor \frac{T}{v} \rfloor$ **for some** $k \in \{1, \ldots, v-1\}$ **then**
4:         Compute patch-wise importance over $\mathcal{P}$ (see Preliminaries section)
5:         $\mathcal{P}_{\text{imp}} \leftarrow$ top-$K$ most important patches in $\mathcal{P}$
6:         $\hat{\mathbf{X}}_k \leftarrow$ synthetic image composed of patches in $\mathcal{P}_{\text{imp}}$
7:         $\mathcal{S} \leftarrow \mathcal{S} \cup \{\hat{\mathbf{X}}_{t_k}\}$                     ▷ Store current sparse image
8:         $\mathcal{P} \leftarrow \mathcal{P} \setminus \mathcal{P}_{\text{imp}}$                     ▷ Detach important patches
9:         $\hat{\mathbf{X}} \leftarrow$ image composed of remaining patches in $\mathcal{P}$
10:     **end if**
11:     Compute $\mathcal{L}_{\text{inv}}(\hat{\mathbf{X}}, y; f)$                             ▷ Eq. (1)
12:     Update $\hat{\mathbf{X}} \leftarrow \hat{\mathbf{X}} - \eta \cdot \nabla_{\hat{\mathbf{X}}} \mathcal{L}_{\text{inv}}$
13: **end for**
14: $\hat{\mathbf{X}}_{t_v} \leftarrow$ final synthetic image with remaining patches in $\mathcal{P}$
15: $\mathcal{S} \leftarrow \mathcal{S} \cup \{\hat{\mathbf{X}}_{t_v}\}$                             ▷ Store final sparse image
16: **return** $\mathcal{S}$

---

Given a pretrained classifier $f$, an initial noise image $\hat{\mathbf{X}}_{t_0}$, and a target label $y$, PRI performs inversion over $T$ iterations. The core idea is to progressively detach the most important patches, based on attention-derived importance scores, at regularly spaced detachment points determined by the division factor $v$. At each detachment point $t_k$ (Line 3), the top-$K$ patches (with $K = \lfloor \frac{N}{v} \rfloor$) are extracted to construct an intermediate sparse image $\hat{\mathbf{X}}_{t_k}$ (Lines 4-6), which is stored in the output set $\mathcal{S}$ (Line 7). The remaining patches continue to be inverted in subsequent iterations, allowing previously unimportant regions to accumulate more class-specific information (Lines 11-12). This process repeats until the final step $t = T$, where the last remaining patches are stored as the final sparse image $\hat{\mathbf{X}}_{t_v}$ (Lines 14-15).

## D FULL EXPERIMENTAL DETAILS

To further evaluate inversion performance across various sparsity levels, we additionally consider the sparsity levels of 50% and 86%. In PRI, these correspond to division factors $v = 2$ and $v = 7$, respectively. For fair comparisons, we adjust the pruning ratios in SMI to match the target sparsity, setting them to (0.16, 0.16, 0.16, 0.16) for 50%, and (0.39, 0.39, 0.39, 0.39) for 86%, while

maintaining the same pruning iteration schedule fixed. For the downstream tasks in Tables 3(a) and 3(b), we employ Kullback-Leibler divergence, a standard objective function in knowledge transfer literature.

In terms of implementation details, to synthesize 128 images with PRI, we invert 64 images for $v = 2$, 32 images for $v = 4$, and 18 images for $v = 7$. Since 128 is not exactly divided by 7, we generate 126 images per batch in the case of $v = 7$. While this yields a slight computational advantage in Table 2, it results in a marginal disadvantage in Tables 3(a) and 3(b). We consider these small differences negligible for the purpose of comparison.

In Table A3, we use 128 images per batch and conduct 120 batches for CIFAR-10 and 1,000 batches for CIFAR-100 and Tiny-ImageNet to ensure sufficient training convergence. Although alternative strategies exist (e.g., training over multiple epochs), we evaluate the quality of inversion by fine-tuning the student model using each inverted image exactly once, isolating the impact of the inversion quality itself on downstream performance.

For training the student model in data-free knowledge transfer experiments, we use SGD optimizer with a learning rate of 0.1, weight decay of 1e-4, and momentum of 0.9. For data-free quantization experiments, we also use SGD with a learning rate of 0.01, keeping the same weight decay and momentum, and use batch size of 128 as same as knowledge transfer experiments. There is no learning rate scheduling applied in both settings.

Following prior work, sparse model inversion (SMI) (Hu et al., 2024), we apply standard data augmentations such as random horizontal flipping and normalization when processing inverted data. For test data, only resizing and normalization are used. In Figures 5 and A1, we use only remaining patches of SMI, not discarded. All experiments, including Table A1, are conducted with a fixed seed, 42 for reproducibility.

To ensure reproducibility and facilitate consistent comparison, we leverage publicly accessible pretrained vision models from widely used libraries such as `timm`. Specifically, we select DeiT/16-Tiny, Small, Base, and a CLIP-based ViT model[2] for visualization tasks. ViT-Base/32 model is employed solely for visualization purposes.

## E  ADDITIONAL EXPERIMENTAL RESULTS

**Extended Results of Experiments.** Tables 2 and 3 in the main text summarize our core findings regarding inversion efficiency, data-free quantization, and data-free knowledge distillation. Here, we present comprehensive results across DeiT-Tiny, Small, and Base models with additional sparsity levels, extending the analyses in Tables A1, A2 and A3.

**Additional Results on Inversion Efficiency.** In Table A1, we examine inversion efficiency across various sparsity levels, highlighting PRI's consistent superiority over SMI and DMI. PRI demonstrates significantly improved throughput, reduced FLOPs, and lower GPU memory usage, particularly at higher sparsity (86%). Specifically, PRI achieves up to 129% throughput improvement, 49% FLOPs reduction, and 66% GPU memory savings compared to SMI. These empirical results strongly align with our theoretical predictions, which anticipated greater efficiency gains with larger division factors $v$.

**Additional Results on Data-Free Quantization.** Table A2 reports data-free quantization results on ImageNet-1k for three DeiT backbones under two bit-width settings (W4/A8 and W8/A8). In DeiT-Tiny, DMI outperforms both SMI and PRI, whereas in DeiT-Base, PRI surpasses DMI despite being $10\times$ faster. PRI consistently achieves superior or competitive accuracy compared to SMI across various sparsity levels.

**Additional Results on Data-Free Knowledge Distillation.** Table A3 provides an extensive validation of PRI's effectiveness in data-free knowledge distillation across CIFAR-10, CIFAR-100, and Tiny-ImageNet datasets. PRI consistently outperforms SMI across all sparsity settings, achieving comparable or superior accuracy to DMI at moderate sparsity levels (50%). In particular, on the CIFAR-10 dataset, PRI with 50% sparsity achieves higher knowledge distillation accuracy and faster

---

[2]`https://huggingface.co/openai/clip-vit-base-patch32`

Table A1: Inversion efficiency on DeiT-Tiny, DeiT-Small, and DeiT-Base across various sparsity levels. Throughput is the inversion speed, measuring to compute inversion iterations per second. The changes in red and blue refer to the comparison with each sparsity level of SMI.

| | | | | | | | |
|---|---|---|---|---|---|---|---|
| Model: **DeiT-Tiny** | | | | | | | |
| Method | DMI | | SMI | | | PRI | |
| Sparsity | 0% | 50% | 76% | 86% | 50% ($v=2$) | 75% ($v=4$) | 86% ($v=7$) |
| Throughput (its/s) ↑ | 6.49 | 10.94 | 15.33 | 18.04 | 15.27 (+39.6%) | 30.71 (+100.3%) | **40.92** (+126.8%) |
| FLOPs (G) ↓ | 949.11 | 454.85 | 229.89 | 143.20 | 348.88 (-23.3%) | 144.09 (-37.3%) | **73.86** (-48.4%) |
| GPU Memory (GB) ↓ | 6.00 | 3.32 | 2.21 | 1.79 | 2.38 (-28.3%) | 1.07 (-51.6%) | **0.60** (-66.5%) |
| Model: **DeiT-Small** | | | | | | | |
| Method | DMI | | SMI | | | PRI | |
| Sparsity | 0% | 50% | 76% | 86% | 50% ($v=2$) | 75% ($v=4$) | 86% ($v=7$) |
| Throughput (its/s) ↑ | 3.05 | 5.75 | 9.10 | 12.14 | 7.72 (+34.2%) | 16.87 (+85.4%) | **27.82** (+129.2%) |
| FLOPs (G) ↓ | 3524.20 | 1729.66 | 881.33 | 545.88 | 1300.98 (-24.8%) | 539.85 (-38.7%) | **277.23** (-49.2%) |
| GPU Memory (GB) ↓ | 11.66 | 5.67 | 3.43 | 2.62 | 4.47 (-21.2%) | 2.01 (-41.4%) | **1.17** (-55.3%) |
| Model: **DeiT-Base** | | | | | | | |
| Method | DMI | | SMI | | | PRI | |
| Sparsity | 0% | 50% | 76% | 86% | 50% ($v=2$) | 75% ($v=4$) | 86% ($v=7$) |
| Throughput (its/s) ↑ | 1.10 | 2.20 | 3.92 | 5.58 | 2.88 (+30.9%) | 6.40 (+63.3%) | **11.81** (+111.6%) |
| FLOPs (T) ↓ | 13.43 | 6.74 | 3.45 | 2.13 | 5.02 (-25.5%) | 2.09 (-39.4%) | **1.07** (-49.8%) |
| GPU Memory (GB) ↓ | 23.42 | 10.77 | 6.26 | 4.61 | 8.99 (-16.5%) | 4.28 (-31.6%) | **2.68** (-41.9%) |

Table A2: Data-free quantization results on ImageNet-1k using different inversion methods across various sparsity levels. W4/A8 refers to the bit precision for weight and activation quantization, respectively. The changes in red and blue refer to the comparison with each sparsity level of SMI. Teacher model accuracies for DeiT-Tiny, Small, and Base are 71.5%, 79.4%, and 81.7%, respectively.

| Dataset: **ImageNet-1k** | | | | | | | | |
|---|---|---|---|---|---|---|---|---|
| Method | | DMI | | SMI | | | PRI | |
| Sparsity | | 0% | 50% | 76% | 86% | 50% ($v=2$) | 75% ($v=4$) | 86% ($v=7$) |
| DeiT-Tiny | W4/A8 | **68.03** | 67.39 | 66.65 | 66.55 | 67.64 (+0.25) | 66.79 (+0.14) | 66.44 (-0.11) |
| | W8/A8 | **69.45** | 68.94 | 68.57 | 68.38 | 69.20 (+0.26) | 68.77 (+0.20) | 68.40 (+0.02) |
| DeiT-Small | W4/A8 | 77.01 | 76.80 | 76.19 | 75.37 | **77.06** (+0.26) | 76.40 (+0.21) | 76.03 (+0.66) |
| | W8/A8 | **78.15** | 78.11 | 77.64 | 77.01 | 77.90 (-0.21) | 77.72 (+0.08) | 77.42 (+0.41) |
| DeiT-Base | W4/A8 | 80.19 | 80.29 | 79.77 | 79.20 | **80.36** (+0.07) | 80.13 (+0.46) | 80.07 (+0.87) |
| | W8/A8 | 80.73 | 80.77 | 80.33 | 79.85 | **80.78** (+0.01) | 80.70 (+0.37) | 80.57 (+0.72) |

inversion speed compared to DMI. Remarkably, for DeiT-Tiny, the accuracy difference between DMI and PRI exceeds 4%. Notably, as sparsity increases (to 75% and 86%), PRI significantly widens its performance gap over SMI, further demonstrating its robustness and efficacy in generating class-agnostic features that are essential for effective knowledge transfer in data-free settings.

**Additional Analyses of Progressive Shift from General to Specific.** We further investigate the role of sparse images generated at different detachment points of PRI by conducting two analyses: one-class distillation in Table A4 and data-free quantization in Table A5.

In the one-class distillation setting in Table A4, we use 2.5k sparse images generated at each of the four detachment points, as well as the combined set denoted as "All" on CIFAR-10. For evaluation, we report the classification accuracy and average KL-divergence loss with respect to the teacher model logits, denoted as KL10 (all classes including airplane) and KL9 (excluding airplane). While training on 2.5k sparse images from every detachment point leads to strong accuracy, we observe that images from the 2nd detachment point achieve the best (i.e., minimum) KL-divergence scores, especially on KL9, suggesting stronger class-agnostic properties. Similarly, the 3rd point images also exhibit favorable generalization, whereas the 4th points show diminished performance but still surpass DMI and SMI in Figure 4.

In the 8-bit data-free quantization experiments on ImageNet-1k in Table A5, we use 2.5k sparse images from each detachment point and the combined set again. We find that images from the 1st detachment point most effectively recover accuracy in quantized models, consistent with their rich class-agnostic feature content. Using all detachment points together also yields strong performance.

Table A3: Data-free knowledge distillation results on CIFAR-10, CIFAR-100, and Tiny-ImageNet different inversion methods across various sparsity levels. The changes in red refer to the comparison with each sparsity level of SMI.

| Dataset: **CIFAR-10** (Teacher: DeiT-Base, Acc: 95.4) | | | | | | | |
|---|---|---|---|---|---|---|---|
| Method | DMI | SMI | | | PRI | | |
| Sparsity | 0% | 50% | 76% | 86% | 50% ($v=2$) | 75% ($v=4$) | 86% ($v=7$) |
| Student Accuracy — DeiT-Tiny | 76.67 | 69.94 | 55.22 | 26.69 | **81.21** (+11.27) | 75.70 (+20.48) | 57.96 (+31.27) |
| Student Accuracy — DeiT-Small | 86.72 | 87.44 | 82.74 | 58.91 | **88.60** (+1.16) | 87.30 (+4.56) | 81.21 (+22.3) |
| Student Accuracy — DeiT-Base | 95.00 | 95.01 | 94.49 | 88.38 | **95.13** (+0.12) | 95.08 (+0.59) | 94.56 (+6.18) |
| Dataset: **CIFAR-100** (Teacher: DeiT-Base, Acc: 80.6) | | | | | | | |
| Method | DMI | SMI | | | PRI | | |
| Sparsity | 0% | 50% | 76% | 86% | 50% ($v=2$) | 75% ($v=4$) | 86% ($v=7$) |
| Student Accuracy — DeiT-Tiny | **54.90** | 48.34 | 24.31 | 3.55 | 54.57 (+6.23) | 43.32 (+19.01) | 21.27 (+17.72) |
| Student Accuracy — DeiT-Small | 67.62 | 62.55 | 45.05 | 11.25 | **67.70** (+5.15) | 62.93 (+17.87) | 45.59 (+34.34) |
| Student Accuracy — DeiT-Base | 79.76 | 79.41 | 77.55 | 70.22 | **79.98** (+0.57) | 79.57 (+1.98) | 78.46 (+8.24) |
| Dataset: **Tiny-ImageNet** (Teacher: DeiT-Base, Acc: 84.6) | | | | | | | |
| Method | DMI | SMI | | | PRI | | |
| Sparsity | 0% | 50% | 76% | 86% | 50% ($v=2$) | 75% ($v=4$) | 86% ($v=7$) |
| Student Accuracy — DeiT-Tiny | **53.98** | 44.96 | 14.01 | 1.97 | 53.55 (+8.59) | 35.65 (+24.64) | 12.37 (+10.4) |
| Student Accuracy — DeiT-Small | 73.42 | 67.21 | 42.51 | 7.60 | **73.52** (+6.31) | 67.50 (+24.99) | 46.51 (+38.91) |
| Student Accuracy — DeiT-Base | **83.95** | 83.51 | 80.62 | 71.55 | 83.85 (+0.34) | 83.68 (+3.06) | 82.47 (+10.92) |

As in the one-class distillation setting, images from the 4th detachment point, which contain the most class-specific features among all detachment points, yield the lowest quantization recovery accuracy.

These results highlight that earlier detachment point images, encoding more class-agnostic knowledge, play a critical role in enhancing both generalization and robustness in data-free learning.

Table A4: One-class distillation results on CIFAR-10 using sparse images inverted at each detachment point of PRI. "All" denotes inverted images using every detachment point.

Table A5: Data-free quantization performance on ImageNet-1k using sparse images inverted at each detachment point. "All" denotes inverted images using every detachment point.

| Dataset: **CIFAR-10** (only "airplane" class) | | | | | |
|---|---|---|---|---|---|
| Detachment Point | All | 1st | 2nd | 3rd | 4th |
| Accuracy (%) ↑ | **93.84** | 93.28 | 93.76 | 93.75 | 90.35 |
| KL10 (×1e-6) ↓ | 8097 | 8242 | **7765** | 7976 | 13602 |
| KL9 (×1e-6) ↓ | 9806 | 9993 | **9411** | 9668 | 16488 |

| Dataset: **ImageNet-1k** (W8/A8) | | | | | |
|---|---|---|---|---|---|
| Detachment Point | All | 1st | 2nd | 3rd | 4th |
| DeiT-Tiny | 67.49 | 67.47 | 67.42 | **67.53** | 67.26 |
| DeiT-Small | 76.20 | **76.29** | 76.15 | 76.21 | 75.66 |
| DeiT-Base | 79.92 | **79.94** | 79.86 | 79.85 | 79.58 |

**t-SNE Visualization.** In Figure A1, we present t-SNE visualizations of the feature embeddings from PRI-inverted images at each detachment point, using DeiT-Base on CIFAR-100. PRI-1st through PRI-4th refer to the t-SNE visualizations of sparse images stored at each progressive detachment stage during inversion.

Unlike class-conditioned visualizations, we assign each sample the label predicted by the teacher model, rather than the originally targeted inversion class. This choice reflects that early point images often exhibit low confidence, as shown in Figure 6(a), indicating that even the teacher struggles to confidently identify them.

Interestingly, despite their low semantic certainty, PRI-1st samples still contribute meaningfully to knowledge transfer, as demonstrated in our analytic results above. Furthermore, we observe a point-wise difference of class separation: PRI-1st embeddings are relatively entangled, gradually becoming more class-discriminative in PRI-2nd and PRI-3rd, and eventually resemble the tightly clustered patterns seen in DMI and SMI at PRI-4th. This progression supports our interpretation that PRI balances class-specific and class-agnostic features over time.

# F  PROOF OF THEOREM 1

In this section, we present the proof of Theorem 1, which theoretically demonstrates that PRI achieves a lower computational cost compared to SMI and DMI in both self-attention (SA) and feed-forward

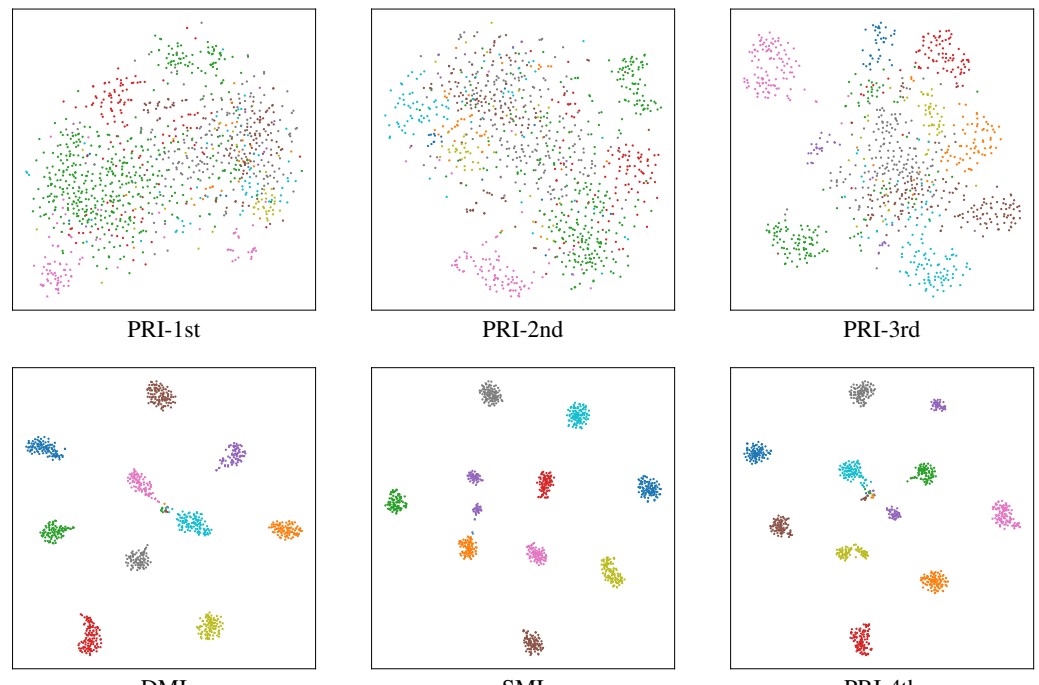

| PRI-1st | PRI-2nd | PRI-3rd |
| --- | --- | --- |
| DMI | SMI | PRI-4th |

Figure A1: t-SNE visualization of feature embeddings from DMI, SMI, and multiple detachment points of PRI on CIFAR-100. All embeddings are extracted using the pretrained DeiT-Base.

network (FFN) modules of ViT-based architectures. Specifically, we show that PRI is more efficient in the SA module under the condition $\frac{N}{d} < 3$, and strictly more efficient in the FFN module regardless of the $\frac{N}{d}$ ratio. This theoretical result aligns with our experimental results, where PRI consistently outperforms other inversion methods across various settings.

*Proof.* To compare the computational costs of different inversion strategies, we focus on the dominant components of the ViT architecture: the self-attention (SA) and feed-forward network (FFN) modules. The per-layer cost of SA is given by $4Nd^2 + 2N^2d$, and the per-layer cost of FFN is approximated as $8Nd^2$ (Chen et al., 2023b). Other components such as LayerNorm and residual connections are omitted as they contribute negligible overhead compared to the main computational terms and do not affect the asymptotic behavior of the comparison.

**Dense Model Inversion (DMI).** Let $N$ be the number of patches per image, $d$ the embedding dimension, $I$ the number of images, $T$ the number of inversion iterations, and $L$ the number of layers. The total computational cost of DMI is:

$$\mathcal{C}_{\text{DMI}}^{\text{SA}} = L \cdot (4Nd^2 + 2N^2d) \cdot I \cdot T, \tag{3}$$

$$\mathcal{C}_{\text{DMI}}^{\text{FFN}} = L \cdot 8Nd^2 \cdot I \cdot T. \tag{4}$$

**Sparse Model Inversion (SMI).** Assuming that SMI produces the same sparsity level as PRI, the output of SMI contains $\frac{N}{v}$ patches, where $v > 1$ is the division factor of PRI. For a fair comparison, we consider an idealized version of SMI in which patch pruning occurs before the inversion process begins, even though the real SMI implementation gradually prunes unimportant patches in the early stages of inversion. By replacing $N$ with $\frac{N}{v}$ in Eqs. equation 3 and equation 4, the computational cost becomes:

$$\mathcal{C}_{\text{SMI}^*}^{\text{SA}} = L \cdot \left( \frac{4Nd^2}{v} + \frac{2N^2d}{v^2} \right) \cdot I \cdot T,$$

$$\mathcal{C}_{\text{SMI}^*}^{\text{FFN}} = L \cdot \frac{8Nd^2}{v} \cdot I \cdot T.$$

**Patch Rebirth Inversion (PRI).** PRI controls the number and timing of patch detachments using a division factor $v$, with each stage running for $\frac{T}{v}$ iterations. Let $N_k$ denote the number of active patches in the $k$-th stage. Since each detachment step removes $\frac{N}{v}$ patches, we have:

$$N_k = N \cdot \left(1 - \frac{k-1}{v}\right), \quad k = 1, \dots, v.$$

Moreover, since PRI generates a group of $v$ sparse images within a single inversion trajectory spanning $T$ iterations, the effective per-image cost should be scaled by a factor of $\frac{1}{v}$. Accordingly, for each stage, only $\frac{I}{v}$ images are effectively counted per synthetic image, and the cost aggregates over all $v$ detachment points through inversion iterations as follows:

$$\mathcal{C}_{\text{PRI}}^{\text{SA}} = \sum_{k=1}^{v} L \cdot \left(4N_k d^2 + 2N_k^2 d\right) \cdot \frac{I}{v} \cdot \frac{T}{v},$$

$$\mathcal{C}_{\text{PRI}}^{\text{FFN}} = \sum_{k=1}^{v} L \cdot 8N_k d^2 \cdot \frac{I}{v} \cdot \frac{T}{v}.$$

Substituting $N_k$ and simplifying yields:

$$\mathcal{C}_{\text{PRI}}^{\text{SA}} = \frac{LIT}{v^2} \sum_{k=1}^{v} \left[4Nd^2\left(1 - \frac{k-1}{v}\right) + 2N^2 d\left(1 - \frac{k-1}{v}\right)^2\right]$$

$$= \frac{LIT}{v^2} \left[4Nd^2 \sum_{k=1}^{v}\left(1 - \frac{k-1}{v}\right) + 2N^2 d \sum_{k=1}^{v}\left(1 - \frac{k-1}{v}\right)^2\right]$$

$$= \frac{LIT}{v^2} \left[4Nd^2 \sum_{j=0}^{v-1}\left(1 - \frac{j}{v}\right) + 2N^2 d \sum_{j=0}^{v-1}\left(1 - \frac{j}{v}\right)^2\right]$$

$$= \frac{LIT}{v^2} \left(\frac{4Nd^2}{v} \sum_{j=1}^{v} j + \frac{2N^2 d}{v^2} \sum_{j=1}^{v} j^2\right)$$

$$= \frac{LIT}{v^2} \left[\frac{4Nd^2}{v} \cdot \frac{v(v+1)}{2} + \frac{2N^2 d}{v^2} \cdot \frac{v(v+1)(2v+1)}{6}\right]$$

$$= \frac{LIT}{v^2} \left[\frac{4Nd^2 \cdot v(v+1)}{2v} + \frac{2N^2 d \cdot v(v+1)(2v+1)}{6v^2}\right],$$

$$\mathcal{C}_{\text{PRI}}^{\text{FFN}} = \frac{LIT}{v^2} \cdot \frac{8Nd^2 \cdot v(v+1)}{2v}.$$

**Feed-Forward Network.** We now compare the computational cost of the feed-forward network (FFN) modules. First, comparing the cost of FFN computations between DMI and SMI yields:

$$\mathcal{C}_{\text{DMI}}^{\text{FFN}} - \mathcal{C}_{\text{SMI}^*}^{\text{FFN}} = L \cdot 8Nd^2 \cdot I \cdot T \left(1 - \frac{1}{v}\right).$$

Since $v > 1$, this difference is always positive, indicating that SMI reduces FFN cost compared to DMI. Moreover, the speedup of SMI over DMI is exactly $v\times$, as shown below:

$$\frac{\mathcal{C}_{\text{DMI}}^{\text{FFN}}}{\mathcal{C}_{\text{SMI}^*}^{\text{FFN}}} = \frac{L \cdot 8Nd^2 \cdot I \cdot T}{L \cdot \frac{8Nd^2}{v} \cdot I \cdot T} = v.$$

Next, we compare the cost of PRI against SMI:

$$\frac{\mathcal{C}_{\text{PRI}}^{\text{FFN}}}{\mathcal{C}_{\text{SMI}^*}^{\text{FFN}}} = \frac{\frac{L \cdot I \cdot T}{v^2} \cdot \frac{8N \cdot d^2 \cdot v(v+1)}{2v}}{L \cdot \frac{8Nd^2}{v} \cdot I \cdot T} = \frac{v+1}{2v} = \frac{1}{2} + \frac{1}{2v}.$$

For the smallest value $v = 2$, PRI is 25% more efficient than SMI in terms of FFN cost. As $v$ increases, this ratio converges to $\frac{1}{2}$, indicating that PRI can become up to $2\times$ more efficient than SMI in the limit.

These highlight a key advantage of PRI: it consistently achieves lower computational cost than both SMI and DMI for FFN computations, independent of the total number of patches and model scalability.

Summarizing:

$$\mathcal{C}_{\text{PRI}}^{\text{FFN}} < \mathcal{C}_{\text{SMI}^*}^{\text{FFN}} < \mathcal{C}_{\text{DMI}}^{\text{FFN}}.$$

**Self-Attention.** We now turn to analyzing the computational costs of self-attention (SA) modules in DMI and SMI. The difference is given by: $\mathcal{C}_{\text{DMI}}^{\text{SA}}$ and $\mathcal{C}_{\text{SMI}^*}^{\text{SA}}$:

$$\mathcal{C}_{\text{DMI}}^{\text{SA}} - \mathcal{C}_{\text{SMI}^*}^{\text{SA}} = L \cdot I \cdot T \cdot \left[ 4Nd^2 \cdot \left( 1 - \frac{1}{v} \right) + 2N^2d \cdot \left( 1 - \frac{1}{v^2} \right) \right].$$

Since $v > 1$, both terms inside the brackets are strictly positive, which confirms that $\mathcal{C}_{\text{DMI}}^{\text{SA}} > \mathcal{C}_{\text{SMI}^*}^{\text{SA}}$ always holds, regardless of model size or patch dimension.

Next, comparing PRI and SMI:

$$\mathcal{C}_{\text{PRI}}^{\text{SA}} - \mathcal{C}_{\text{SMI}^*}^{\text{SA}} = L \cdot I \cdot T \cdot \left[ \frac{2N^2d}{6v^3}(2v^2 - 3v + 1) - \frac{4Nd^2}{2v^2}(v - 1) \right].$$

Solving the inequality for PRI to be more efficient than SMI gives:

$$N < \frac{6v(v - 1)}{2v^2 - 3v + 1} \cdot d.$$

This bound decreases monotonically with $v$: for $v = 2$, the bound is $N < 4d$; for $v = 3$, it becomes $N < 3.6d$; and as $v \to \infty$, it approaches $N < 3d$. Therefore, in practical regimes where ViTs typically satisfy $N < 3d$, PRI achieves lower self-attention cost compared to both SMI and DMI. This ratio condition $\frac{N}{d} < 3$ is satisfied by most of the standard ViT architectures, including DeiT-Tiny, Small, and Base, where $N = 197$ and $d = 192$, 384, and 768, respectively. It also holds for larger models such as ViT-Large ($N = 197$ and $d = 1024$) and ViT-Huge ($N = 257$ and $d = 1280$).

Summarizing:

$$\mathcal{C}_{\text{PRI}}^{\text{SA}} < \mathcal{C}_{\text{SMI}^*}^{\text{SA}} < \mathcal{C}_{\text{DMI}}^{\text{SA}} \quad \text{(when } \tfrac{N}{d} < 3\text{)}.$$

$\square$

# G  VISUALIZATION DETAILS

Our visualization strategies of model inversion are based on the prior work (Hu et al., 2024). All images shown in Figures 1(a), 2, 3, and A2 are inverted using the CLIP-based ViT/32-Base model as its features have been found to align more closely with human perception due to large-scale pretraining (Hu et al., 2024). For improved visual clarity, we follow the approach of prior work (Hatamizadeh et al., 2022) and incorporate batch normalization borrowed from convolutional neural networks (CNNs). All visualization images are from the CIFAR-100 dataset.

In our empirical observations, fine-tuning the entire model improves the quantitative performance of the pretrained teacher, but does not lead to noticeable improvements in visual quality from a human perception perspective. Instead, we find that fine-tuning only the classifier head yields the best results for visualization purposes. All other experimental settings remain consistent with those used in the knowledge transfer inversion experiments.

# H  VISUALIZATIONS OF THE RE-BIRTH EFFECT

Figure A2 illustrates the re-birth effect across 12 classes on CIFAR-100. The images are arranged in a 4×3 grid, with rows ordered from left-to-right, top-to-bottom by class: *pear, rose, apple, orange, orchid, lion, sunflower, aquarium fish, bus, bee, poppy, and boy*. The emergence of semantic structure in the right column highlights PRI's ability to transform initially uninformative regions into meaningful content through progressive optimization.

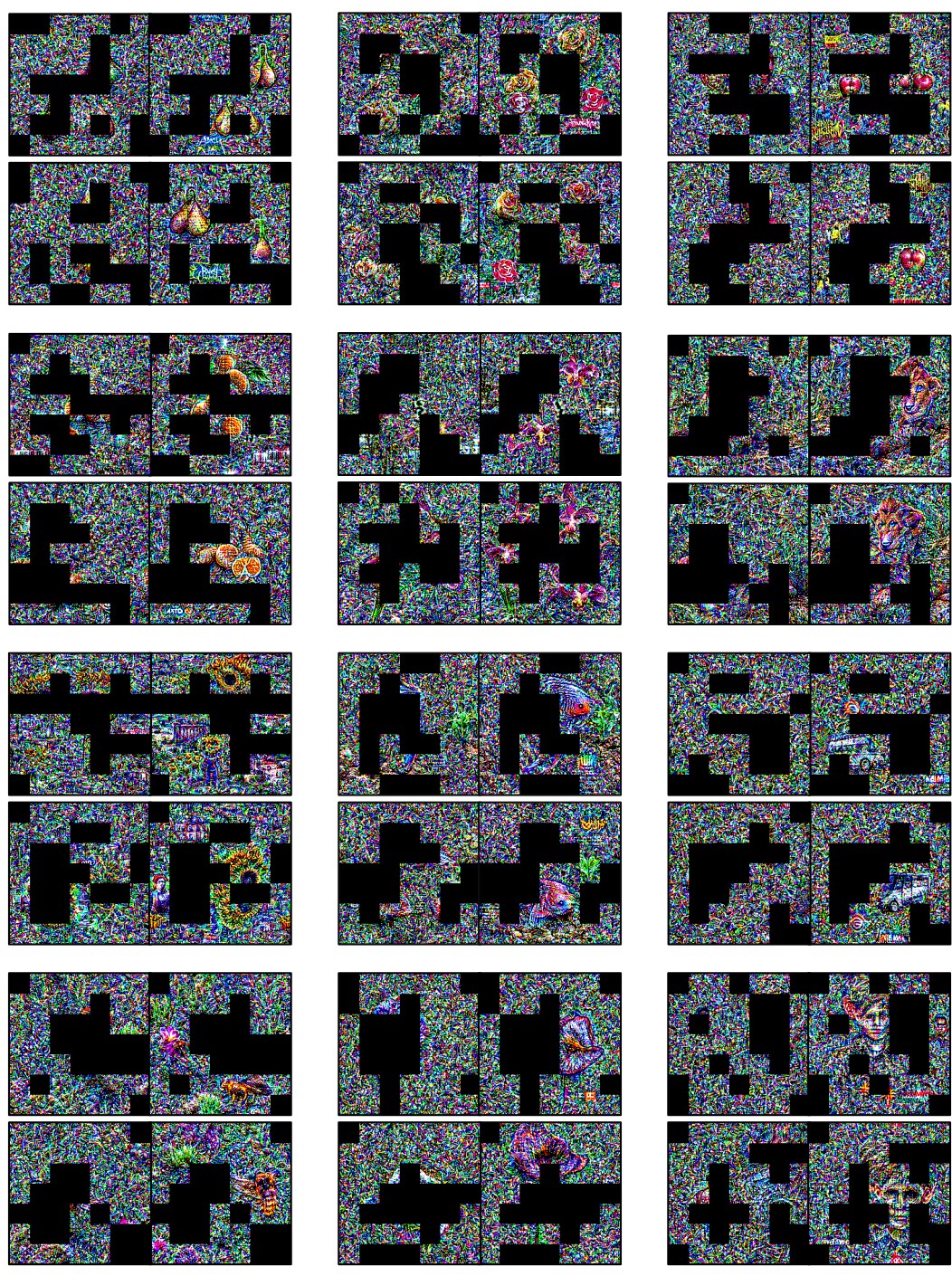

Figure A2: Re-birth visualizations. For each class, the left image shows the initially regarded as unimportant patches, while the right image shows the same patches after further inversion.

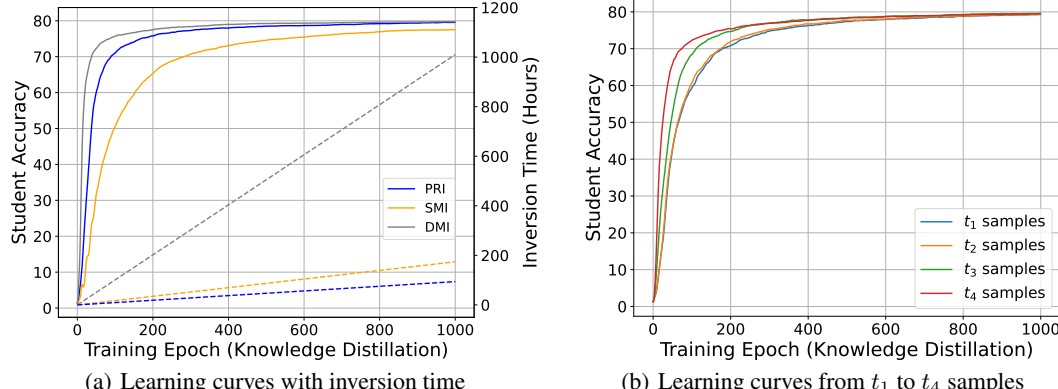

(a) Learning curves with inversion time        (b) Learning curves from $t_1$ to $t_4$ samples

Figure A3: Learning curves on CIFAR-100 using a DeiT-Base teacher and a DeiT-Base student. (a) Comparison of learning curves with inversion time among three inversion methods (PRI with 75% sparsity, SMI with 76% sparsity and DMI). (b) Learning curves using samples generated at different detachment points from $t_1$ to $t_4$.

## I    LEARNING DYNAMICS OF KNOWLEDGE TRANSFERRING

Figure A3 presents a unified analysis of the convergence behavior of PRI, SMI, and DMI, as well as the influence of samples generated at different stages of PRI.

**Convergence Behavior of PRI, SMI, and DMI.** In Figure A3(a), we plot the learning curves of the three methods using the DeiT-Base teacher–student pair on CIFAR-100 (as in Table 3(b)), along with the associated inversion-time requirements. PRI not only converges substantially faster than SMI, but also maintains a clear performance gap even after both methods reach their converged regimes. As expected, DMI achieves the strongest performance throughout the entire trajectory; however, this comes at an extremely high inversion cost (approximately 1,000 hours in this experiment). Overall, the relative ordering remains stable, i.e., DMI $\gtrsim$ PRI > SMI, and the gaps do not diminish over time. This suggests that the differences among the sparsification strategies are not transient optimization artifacts, but rather persist throughout the full training horizon.

**Unified Convergence across Detachment Points.** Figure A3(b) presents the learning curves obtained from samples at four detachment points, $t_1$ through $t_4$, under PRI with division factor $v = 4$. As in Figure A3(a), we use the DeiT-Base teacher–student pair on CIFAR-100. The samples detached at earlier stages ($t_1$ and $t_2$) exhibit nearly identical trajectories, while those from the latest stage ($t_4$) converge more quickly. Nevertheless, all four cases eventually achieve similar final accuracy.

At a glance, this may seem to imply that the later-stage, more class-specific $t_4$ samples are universally advantageous. However, Tables A4 and A5 show that relying solely on $t_4$ samples yields noticeably poorer performance in other settings. This contrast suggests that different detachment points capture complementary properties: early-stage samples retain class-agnostic structures beneficial for stability and robustness, whereas later-stage samples provide class-specific details. Therefore, leveraging samples across multiple detachment points is essential for consistently strong performance in downstream tasks.

## J    THEORETICAL VIEW ON PERFORMANCE DIFFERENCES

To better interpret the performance differences observed among PRI, SMI, and DMI across all experiments, we draw on a recent theoretical analysis from Wei et al. (2025), restated in Theorem A1 below.

**Theorem A1.** *(Wei et al. (2025)) Given the original dataset $\mathcal{D} = \{\boldsymbol{x}_i, y_i\}_{i \in [m]}$ with $m$ i.i.d. samples and the synthetic dataset $\mathcal{S} = \{\hat{\boldsymbol{x}}_j, \hat{y}_j\}_{j \in [s]}$. Assume the hypothesis function is $\lambda^\eta$-Lipschitz continuous, the loss function $\ell(\boldsymbol{x}, y)$ is $\lambda^\ell$-Lipschitz continuous for all $y$, and is bounded by $L$, with*

$\ell(\hat{\boldsymbol{x}}_j, \hat{y}_j; \boldsymbol{\theta}) = 0$ *for all* $j \in [s]$. *If the dataset* $\mathcal{S}$ *is a* $\delta$*-cover of* $\mathcal{D}$*, with probability at least* $1 - \gamma$*, the bound holds:*

$$\left| \frac{1}{m} \sum_{i \in [m]} \ell(\boldsymbol{x}_i, y_i; \boldsymbol{\theta}) - \frac{1}{s} \sum_{j \in [s]} \ell(\hat{\boldsymbol{x}}_j, \hat{y}_j; \boldsymbol{\theta}) \right| \leq \frac{\lambda^\ell + \lambda^\eta LC}{\delta_{div}} + \sqrt{\frac{\log |\boldsymbol{\Theta}| + \log \frac{1}{\gamma}}{2m}},$$

*where* $C$ *is the number of classes, and* $\boldsymbol{\theta} \in \boldsymbol{\Theta}$ *is the optimized student model.*

The theorem states that the generalization error between the true dataset $\mathcal{D}$ and the synthetic dataset $\mathcal{S}$ is fundamentally governed by the diversity of $\mathcal{S}$, quantified by $\delta_{div}$. A larger $\delta_{div}$ (corresponding to greater synthetic-data diversity) tightens the bound and thereby improves the downstream generalization performance of the student model.

A direct implication for our setting is that the performance differences among PRI, SMI, and DMI can be attributed to how diverse their synthesized datasets are. PRI yields synthetic samples with substantially higher diversity due to its progressive sparse reconstruction, leading to a larger $\delta_{div}$ and thus a tighter generalization bound. In contrast, SMI produces less diverse samples because a significant portion of information is removed early in the process, resulting in a looser bound. DMI attains high diversity but only at an impractically high inversion cost, making such diversity unattainable under realistic compute budgets. This diversity-based perspective explains the consistent performance ordering observed across our experiments. Also, this observation is aligned with the qualitative evidence presented in Figures 5 and 6, where PRI consistently produces more diverse synthetic images and smoother, less overconfident confidence distributions than SMI, indirectly reflecting its larger $\delta_{div}$.