# OpenReview forum: "Patch Rebirth: Toward Fast and Transferable Model Inversion of Vision Transformers"
_ICLR.cc/2026/Conference — Submitted to ICLR 2026_

### Official Review · Reviewer_UkW6 · 2025-10-24

**Soundness:** 3
**Presentation:** 2
**Contribution:** 3
**Rating:** 6
**Confidence:** 3

**Summary:**

The authors proposed PRI (Patch Rebirth Inversion): During the inversion process of the ViT model, the most important patches are extracted in stages to form a sparse image and saved, and the remaining patches are continued to be inverted for gradual rebirth, so that one inversion trajectory can generate multiple sparse samples, taking into account both efficiency and transferability; in the two major tasks of quantization and distillation, it is faster and more accurate than SMI and closer to DMI

**Strengths:**

1. A novel and simple approach: using attention scores to estimate patch importance, but rather than discarding low-scoring patches, PRI performs potential rebirth (further optimization) to ultimately produce multiple, non-overlapping, sparse images. The implementation includes clear pseudocode.

2. Excellent efficiency and effectiveness:

   - Efficiency: On DeiT, PRI improves throughput by up to 2 $\times$ compared to SMI and up to 10 $\times$ compared to DMI; it also significantly reduces FLOPS and memory usage.

   - Quantization and Distillation: Quantization Attempts (QAT) outperform DMI at moderate sparsity and SMI at high sparsity; distillation outperforms SMI overall, approaching DMI.

3. Strong evidence of transferability: Students in single-class inversion experiments generalize to all classes under PRI; the confidence distribution is closer to real data, showing a trend from class-independent to class-dependent.

4. Thorough theoretical analysis: A cost comparison between SA and FFN is presented, demonstrating that under the common ViT setting of $$N/D<3$$, PRI outperforms both SA and FFN, as well as DMI and SMI.

**Weaknesses:**

1. When comparing complexity, the SMI was set to an ideal state (assuming pruning was completed once before inversion), which is slightly different from the actual situation.

2. The article uses attention scores as the sole criterion for determining important patches. Are there other more robust criterion?

3. Why is it that after removing important patches, the remaining patches are more likely to synthesize class-related features? Is there any further research on this?

4. Typo: Line 363 should indicate GPU usage, not usuage.

**Questions:**

See Weakness

---

> ### Author Response · Authors · 2025-11-15
> **Response to (W1): Complexity of SMI**
>
> ## **Reviewer's concern**
> > When comparing complexity, the SMI was set to an ideal state (assuming pruning was completed once before inversion), which is slightly different from the actual situation.
>
> ## **Response**
>
>
> Thank you for pointing this out.
> The complexity analysis in Section 4.3 indeed adopts an idealized version of SMI, where patch pruning is assumed to occur once before inversion begins. This assumption intentionally places SMI in a more favorable computational scenario. Even under this idealized setting, however, **PRI still achieves higher efficiency**, indicating that our advantage does not rely on disadvantaging the baseline.

---

> ### Author Response · Authors · 2025-11-15
> **Response to (W2) Patch importance criterion**
>
> ## **Reviewer's concern**
> > The article uses attention scores as the sole criterion for determining important patches. Are there other more robust criterion?
>
> ## **Response**
>
> As described in the paper, we use attention scores as the importance criterion because they are one of the most widely validated and stable proxies for spatial relevance in ViT-based architectures. Prior work on SMI also adopts this choice, and attention-based importance is consistent with the inductive biases of transformer models.
>
> That said, the transformer literature provides alternative criteria, such as per-patch gradient magnitude [1]. To assess robustness, we evaluated PRI with this alternative metric:
> | Setting        | PRI (attention) | PRI (gradient) |
> | -------------- | --------------- | -------------- |
> | DeiT-S $\rightarrow$ Tiny  | 93.23           | 93.33          |
> | DeiT-S $\rightarrow$ Small | 64.77           | 66.68          |
>
> The results show that the two choices yield very similar performance, indicating that PRI is not sensitive to the particular importance metric used.
>
> [1] Generic Attention-model Explainability for Interpreting Bi-Modal and Encoder-Decoder Transformers, ICCV 2021.

---

> ### Author Response · Authors · 2025-11-15
> **Response to (W3): Why remaining patches become more class-related**
>
> ## **Reviewer's concern**
> > Why is it that after removing important patches, the remaining patches are more likely to synthesize class-related features? Is there any further research on this?
>
>
> ## **Response**
>
> Thank you for the very interesting question that touches on the core intuition behind our work.
>
> To clarify, the remaining patches do **not** become class-related immediately after the important patches are removed. Their class relevance emerges gradually through continued inversion. Once high-importance patches are detached, the semantic gradients are effectively re-directed toward the surviving tokens, allowing these patches to acquire class-specific structure over time. This is what underlies the *re-birth* effect.
>
> Comparing $t_1$ and $t_4$ samples:
> - $t_1$ samples have undergone only a short portion of inversion, so they predominantly encode class-agnostic patterns.
> - $t_4$ samples have been optimized over the entire trajectory (when $v=4$), giving them enough time to accumulate class-specific information.
>
> Thus, the remaining patches are not inherently superior; rather, they become class-relevant because PRI allows them to keep evolving instead of being permanently discarded, as in SMI.

---

> ### Author Response · Authors · 2025-11-15
> **Response to (W4): Typos**
>
> ## **Reviewer's concern**
> > Typo: Line 363 should indicate GPU usage, not usuage.
>
> ## **Response**
>
> Thank you! We correct "usuage" to "usage" in the revised version.

---

### Official Review · Reviewer_pfYh · 2025-10-27

**Soundness:** 3
**Presentation:** 3
**Contribution:** 3
**Rating:** 4
**Confidence:** 3

**Summary:**

This paper aims to improve the efficiency of reconstructing synthetic inputs from a pretrained model (i.e., model inversion). Specifically, the authors prune and discard unimportant patches for acceleration and they propose Patch Rebirth Inversion (PRI) that incrementally detaches the most important patches. Experimental results show that PRI achieves much faster inversion than standard Dense Model Inversion (DMI).

**Strengths:**

1. This paper is well-written and easy to follow.
2. The authors reveal the phenomenon that even randomly selected patches can eventually acquire transferable knowledge through continued inversion.
3. The proposed PRI method achieves faster inversion than both dense inversion and previous sparse inversion methods.

**Weaknesses:**

Please refer to Questions.

**Questions:**

Training efficiency and accuracy are difficult to balance, and I have several questions regarding this:

1. Regardless of the sparsity level or sparsity method used during training, if training time is sufficiently long, will the final student model accuracy always converge to the same level? In other words, if we disregard training time, do different sparsification methods fundamentally affect the ultimate accuracy?

2. For both SMI and PRI, if we fix the total training time, could we achieve better performance by training at a higher sparsity level (e.g., 75%) for three times longer compared to a lower sparsity level (e.g., 25%)? Under such a constraint, what is the optimal trade-off between sparsity level and training duration? Specifically, can increasing the sparsity level—thereby allowing more training iterations within the same time budget—lead to higher final accuracy?

3. In Table 3(b), the advantage of PRI over SMI appears more pronounced than in Table 3(a). What causes this difference? For SMI in Table 3(b), would further increasing the training time improve its accuracy? If not, why not? If yes, approximately how much more training time would be needed for SMI to reach the same accuracy as PRI?

---

> ### Author Response · Authors · 2025-11-15
> **Response to (Q1): Do different sparsification methods converge to the same accuracy given sufficient training time?**
>
> ## **Reviewer's concern**
> > Regardless of the sparsity level or sparsity method used during training, if training time is sufficiently long, will the final student model accuracy always converge to the same level? In other words, if we disregard training time, do different sparsification methods fundamentally affect the ultimate accuracy?
>
> ## **Response**
>
> We appreciate the reviewer’s question, as it probes whether the differences among sparsification methods are transient (optimisation-related) or fundamental (data-related). Our experiments show that **even with substantial increases in either inversion time or student-training time, the methods do not converge to the same final accuracy**.
>
> **(1) Extended inversion experiments: increasing inversion iterations does not close the gap.**
> To directly test whether longer inversion eventually eliminates the performance difference, we increased inversion iterations from our standard 4k to 8k and 12k, keeping all other hyperparameters fixed. We used the DeiT-Small $\rightarrow$ DeiT-Tiny distillation setup on CIFAR-10.
>
> | Inversion Iterations | DMI   | SMI   | PRI   |
> | -------------------- | ----- | ----- | ----- |
> | 4,000     | 69.58 | 54.87 | 64.77 |
> | 8,000                | 69.59 | 56.87 | 62.05 |
> | 12,000               | 70.66 | 51.57 | 62.66 |
>
> Although the absolute numbers change slightly with more inversion steps, the relative ordering remains stable: $DMI$ (upper bound) $> PRI > SMI$.
>
> Crucially, SMI never closes the gap to PRI. This confirms that extending inversion time does not cause the different sparsification methods to converge to the same accuracy.
>
>
> **(2) Student-training experiments: long training also does not eliminate the gap.**
>
> In Appendix Figure A3(a) (the revised version), we provide detailed convergence curves for student training over 1,000 epochs. These curves show that both PRI and SMI reach clear convergence within this training horizon: their accuracies plateau well before 1,000 epochs, and extending training further would not improve either method.
>
> Importantly, even though both methods fully converge within this range, PRI consistently converges to a higher accuracy, while SMI plateaus lower. This confirms that the performance gap is not caused by insufficient optimization, but reflects intrinsic differences in the synthetic data produced by the two sparsification strategies.
>
> Thus, examining the full convergence trajectory verifies that longer student training does not eliminate the gap, because both methods have already reached their respective optima.
>
>
> **(3) Why the gaps persist: sparsification differences change the information content**
>
> Training time governs how well the student fits the synthetic dataset, but cannot compensate for differences in the diversity and representational richness of the synthetic data itself. According to Theorem 4.1 of Wei et al. (2025):
>
> $Generalisation~Error \propto \frac{1}{\delta_{div}}$,
> where $\delta_{div}$ quantifies the diversity of the synthetic samples. In this sense, we infer that:
> - SMI discards many patches early, permanently lowering $\delta_{div}$.
> - PRI maintains and progressively refines diverse patches through patch rebirth, yielding higher $\delta_{div}$.
> - DMI preserves full information but at much higher inversion cost.
>
> Because these methods produce synthetic datasets with intrinsically different diversity, their final student accuracy cannot converge to the same value, even with unlimited training or longer inversion. This new theoretical view has been added in the Appendix J.

---

> ### Author Response · Authors · 2025-11-15
> **Response to (Q2): Sparsity vs. Extended Training**
>
> ## **Reviewer's concern**
> > For both SMI and PRI, if we fix the total training time, could we achieve better performance by training at a higher sparsity level (e.g., 75%) for three times longer compared to a lower sparsity level (e.g., 25%)? Under such a constraint, what is the optimal trade-off between sparsity level and training duration? Specifically, can increasing the sparsity level—thereby allowing more training iterations within the same time budget—lead to higher final accuracy?
>
> ## **Response**
>
> Thank you for the reviewer's question on the relationship between sparsity level and training time.
>
> To directly address the reviewer’s question, we compared models trained with:
> - **25% sparsity** under the standard inversion budget (**4,000 iterations**), and
> - **75% sparsity** trained for **3$\times$ longer** (**12,000 inversion iterations**), while keeping all other settings identical.
>
> The results are summarized below.
>
> | Setting                              | Accuracy  |
> | ------------------------------------ | --------- |
> | 25% sparsity SMI, 4k inversion       | **71.44** |
> | 76% sparsity SMI, 4k inversion       | 54.87     |
> | 76% sparsity SMI, **12k inversion (3×)** | 51.57     |
>
> Even when trained for three times longer, the high-sparsity model (76%) performs substantially worse than the low-sparsity model (25%), despite the latter using only one-third of the inversion iterations.
>
> This demonstrates that, under a fixed training-time or compute budget, **preserving more visual information (i.e., lower sparsity) is generally more beneficial than extending the inversion duration**. Longer optimization cannot compensate for the loss of patch-level information at high sparsity.
>
> This pattern is also consistent with the widely observed gap between dense inversion (DMI) and sparse inversion (SMI):
> methods that retain richer patch information consistently yield better synthetic data, and additional training time cannot recover the information lost through aggressive sparsification.

---

> ### Author Response · Authors · 2025-11-15
> **Response to (Q3): Why the PRI–SMI gap is larger in Table 3(b) than in Table 3(a)**
>
> ## **Reviewer's concern**
> > In Table 3(b), the advantage of PRI over SMI appears more pronounced than in Table 3(a). What causes this difference? For SMI in Table 3(b), would further increasing the training time improve its accuracy? If not, why not? If yes, approximately how much more training time would be needed for SMI to reach the same accuracy as PRI?
>
>
> ## **Response**
>
> The more pronounced advantage of PRI over SMI in Table 3(b) originates from the fundamentally different difficulty levels of the two tasks.
>
> **(1) Distillation (Table 3(b)) is a much harsher scenario than quantisation (Table 3(a)).**
>
> In Table 3(b), the student experiences a large accuracy drop during distillation:
> - DeiT-Base $\rightarrow$ DeiT-Tiny: **80.6%** $\rightarrow$ **54.9%**
>
> This severe loss of information leaves substantial room for improvement, allowing differences between sparsification strategies to be more visible.
>
> In contrast, the quantization setting in Table 3(a) starts with very small degradation:
> - Full-precision: **81.7%**
> - 8-bit QAT (with DMI): **80.73%**
>
> Because the baseline already preserves most information, any method (SMI, PRI, DMI) can show only modest improvements.
>
>
> **(2) Increasing training time for SMI does not close the gap.**
>
> As mentioned in our response to (Q1), extended inversion experiments (4k $\rightarrow$ 8k $\rightarrow$ 12k iterations) show that:
> - SMI’s accuracy fluctuates slightly with longer inversion,
> - but **never approaches PRI's accuracy**,
> - and may even degrade at very high iteration counts.
>
> The reason is structural, not computational:
> - In SMI, once patches are discarded, they receive **zero updates**.
> - The **representational capacity lost** at early pruning **cannot be recovered**, regardless of how long training continues.
> - Therefore, no practical amount of extra training can enable SMI to match PRI or DMI.

---

### Official Review · Reviewer_UtL8 · 2025-10-28

**Soundness:** 3
**Presentation:** 3
**Contribution:** 2
**Rating:** 6
**Confidence:** 4

**Summary:**

This paper addresses the problem of model inversion for vision transformer (ViT) architectures in data-free settings. Their main contributions are:

* The authors observe that while model inversion (i.e., reconstructing synthetic inputs from a pretrained model without original training data) is well studied for CNNs, applying it to modern ViTs is challenging due to expensive self-attention layers and patch-based architecture. They note that prior works (e.g., sparse model inversion) prune away patches during inversion assuming they are unimportant, but this may hinder transferability.

* The authors show that patches which seem unimportant early in inversion can later acquire transferable knowledge if allowed to continue evolving (rather than being discarded). In other words, prematurely discarding patches reduces the ability to capture class-agnostic and class-specific features.

* They propose a new inversion algorithm that incrementally detaches the most “important” patches during inversion (i.e., progressively freeze or remove them) but allows the remaining patches to continue optimizing (“rebirth”) and accumulate knowledge for future selection. This balances between focusing on high-utility patches and giving less-useful ones a chance to evolve.

* They demonstrate empirically that PRI achieves significant speed-ups (e.g., up to ~10× faster than standard Dense Model Inversion, and ~2× faster than prior Sparse Model Inversion) while maintaining or improving inversion quality (in terms of downstream transferability of the synthetic data) on ViTs.

**Strengths:**

* As vision transformers become ubiquitous, methods for data-free learning, model inversion, and knowledge transfer for ViTs are increasingly important. This paper addresses a timely gap.


* The “Re-Birth” effect of patches is an interesting observation, challenging the assumption that low-importance patches early on can simply be discarded. This insight adds value beyond just proposing another method.


* The design of PRI — progressively detaching important patches, allowing the rest to evolve — is well motivated and intuitively appealing for patch-based architectures.


* The reported speed-ups (10× vs. dense inversion, 2× vs. sparse inversion) plus maintenance or improvement of transfer performance is a compelling result that suggests real practical value.

**Weaknesses:**

* The method relies on identifying which patches are “important” and detaching them during the process. It would help to see more detailed justification of how importance is measured, how thresholds are set, and how robust this is across architectures/datasets.


* The experiments appear focused on vision transformers; it is less clear how well the method would extend to other modalities (e.g., NLP transformers) or hybrid architectures. The paper could elaborate on this.


* While the Re-Birth effect is interesting, it would be beneficial to provide more analysis/visualisation of how patches evolve, what kinds of features they capture, and how that relates to transferability. Without that, the claim is less grounded.

**Questions:**

* How exactly do you measure patch “importance” in your method? Is it based on attention scores, gradient magnitude, feature magnitude, or another metric? How sensitive is the performance to the choice of this metric?


* What are the hyperparameters governing when and how many patches to detach in each iteration? Did you perform ablation studies on these choices? How do different schedules (e.g., detach early vs later) affect both speed and quality?


* What downstream tasks did you evaluate the synthetic images on (e.g., fine‐tuning, zero‐shot, domain adaptation)? How much of the transfer gain comes from class‐agnostic features vs. class‐specific features?

---

> ### Author Response · Authors · 2025-11-15
> **Response to (W1) and (Q1): Patch importance metric**
>
> ## **Reviewer's concern**
> > The method relies on identifying which patches are “important” and detaching them during the process. It would help to see more detailed justification of how importance is measured, how thresholds are set, and how robust this is across architectures/datasets.
>
> > How exactly do you measure patch “importance” in your method? Is it based on attention scores, gradient magnitude, feature magnitude, or another metric? How sensitive is the performance to the choice of this metric?
>
> ## **Response**
>
> Thank you for raising this point.
>
> **Patch Importance Metric**
>
> As described in the Preliminaries section, we measure patch importance using attention scores, which is also the standard choice in prior SMI work and a natural signal for spatial relevance in ViTs. Our detachment procedure simply selects the **top-K** highest-importance patches without a particular threshold.
>
> **Datasets and architectures**
>
> Our experiments already span multiple architectures and datasets (CIFAR-10/100, TinyImageNet; DeiT-Tiny/Small/Base), and consistently show stable behaviour, indicating that this metric is sufficiently robust in practice.
>
> **Gradient-based metric**
>
> To further examine sensitivity of importance metrics, we also conducted an experiment using a **gradient-based** importance metric:
> | Setting            | PRI (attention) | PRI (gradient) |
> | ------------------ | -------------- | -------------- |
> | DeiT-Small $\rightarrow$ Tiny  | 93.23          | 93.33          |
> | DeiT-Small $\rightarrow$ Small | 64.77          | 66.68          |
>
> The results are roughly identical, showing that PRI is **not sensitive to the particular choice of importance metric** and remains effective under alternative criteria.

---

> ### Author Response · Authors · 2025-11-15
> **Response to (W2): Generalisation beyond ViTs**
>
> ## **Reviewer's concern**
> > The experiments appear focused on vision transformers; it is less clear how well the method would extend to other modalities (e.g., NLP transformers) or hybrid architectures. The paper could elaborate on this.
>
>
> ## **Response**
>
> We agree that understanding how PRI may extend beyond vision transformers is valuable, and we appreciate the reviewer’s suggestion to discuss this more explicitly.
>
> Our current work focuses on ViT-based architectures because patch-level inversion and attention-based importance estimation align naturally with fixed-size visual tokens. This makes it straightforward to detach and evolve token subsets during inversion.
>
> That said, we believe that the core principle of PRI has the potential to generalise to other transformer modalities. Applying PRI to NLP or hybrid models, however, would require additional design considerations, such as handling variable-length token sequences.
>
> We think of this as a promising direction for future research, and we thank the reviewer for highlighting it.

---

> ### Author Response · Authors · 2025-11-15
> **Response to (W3): Analysis of the re-birth effect**
>
> ## **Reviewer's concern**
> > While the Re-Birth effect is interesting, it would be beneficial to provide more analysis/visualisation of how patches evolve, what kinds of features they capture, and how that relates to transferability. Without that, the claim is less grounded.
>
> ## **Response**
> Thank you for highlighting the Re-Birth effect as an interesting aspect of our work and for suggesting a deeper analysis on it. We appreciate this suggestion, as understanding how patches evolve is central to the motivation behind PRI.
>
> In the main paper, Section 5 provides a detailed examination of the feature evolution across the detachment stages $t_1$ to $t_4$, analysing how their learned representations differ and how these differences influence downstream transferability. To further strengthen this claim:
> - **Appendix E** provides an extended set of quantitative and qualitative analyses, showing how different detachment stages encode complementary strengths (Tables A3, A4, and A5).
> - **Appendix E** also includes t-SNE visualisations (Figure A1) illustrating the progression of feature embeddings from entangled, low-certainty representations in early stages to more sharply clustered, class-discriminative patterns in later stages.
> - **Appendix H** includes visualisations illustrating the Re-Birth effect, showing how initially low-importance patches eventually acquire semantically meaningful structure during inversion.

---

> ### Author Response · Authors · 2025-11-15
> **Response to (Q2): Detachment schedule hyperparameters**
>
> ## **Reviewer's concern**
> > What are the hyperparameters governing when and how many patches to detach in each iteration? Did you perform ablation studies on these choices? How do different schedules (e.g., detach early vs later) affect both speed and quality?
>
> ## **Response**
>
> Thank you for raising this point about the detachment schedule, which indeed touches on a possible extension of PRI.
>
> As described in Section 4.2, the detachment schedule is governed by the total inversion steps $T$ and the division factor $v$, which define a sequence of detachment points $t_1,\ldots,t_v$. In our PRI method, we adopt a uniform schedule, dividing the inversion trajectory into $v$ equal segments. Thus, the interval between $t_k$ and $t_{k+1}$ remains the same (i.e., $T/v$) for every $k \in [1, v-1]$.
>
> This choice is motivated by an inherent trade-off:
> - **Detaching too early** overly reduces the number of active patches, which limits the accumulation of class-related knowledge.
> - **Detaching too late** preserves more knowledge but sacrifices efficiency, reducing the overall benefit of sparse inversion.
> - A **uniform schedule** retains sufficient diversity and generality while maintaining the computational advantages of sparsity.
>
> We selected the uniform schedule precisely because it sits at this balanced middle point, offering stable performance across architectures and datasets without requiring heavy hyperparameter tuning.
>
> While it is possible to explore non-uniform schedules (e.g., front-skewed or back-skewed detachment), our goal in this work was to design a method that is simple, reproducible, and robust across diverse settings. Given that **Section 5 and Appendix E already demonstrate how each detachment stage contributes complementary knowledge**, we regard *fine-grained control of the detachment timing* as an interesting direction for future extensions of PRI.

---

> ### Author Response · Authors · 2025-11-15
> **Response to (Q3): Downstream tasks and feature contributions**
>
> ## **Reviewer's concern**
> > What downstream tasks did you evaluate the synthetic images on (e.g., fine‐tuning, zero‐shot, domain adaptation)? How much of the transfer gain comes from class‐agnostic features vs. class‐specific features?
>
> ## **Response**
> Thank you for this question regarding the downstream tasks and the roles of class-agnostic and class-specific features.
>
> **Downstream tasks**
>
> Our experiments focus on standard supervised classification, which is the primary downstream scenario relevant to our objectives. Specifically, we evaluate synthetic images on two representative data-free learning tasks:
> - Knowledge Distillation (KD)
> - Quantization-Aware Training (QAT)
>
> These are the two canonical settings in prior model inversion literature, and they directly reflect the intended use case of the synthetic data produced by PRI.
>
> **Class-agnostic vs. class-specific features**
>
> Regarding the contribution of class-agnostic versus class-specific features, our analyses in Section 5, Appendix E, and Appendix I show that both components are necessary and complementary:
> - As shown in Figure A3(b), models trained solely on $t_4$ samples, which represent later-stage, class-specific patches, still converge rapidly, indicating that reborn patches do acquire strong semantic relevance.
> - However, this effect does not generalize uniformly. For example, in data-free quantization (Table A5), $t_4$ samples alone yield the lowest recovery performance, demonstrating insufficient class-agnostic coverage.
> - Likewise, early-stage samples ($t_1$) contain mostly class-agnostic information and perform well in certain settings (e.g., one-class distillation), but they cannot match the full mixture.
>
> Together, these results highlight that neither early-stage (class-agnostic) nor late-stage (class-specific) samples alone are sufficient.
> The full mixture ("All") across detachment stages is critical for achieving consistently strong transferability across KD and QAT tasks.

---

> ### Author Response · Authors · 2025-11-16
> **Ablation study on alternative detachment schedules**
>
> ## **Reviewer's concern**
> > What are the hyperparameters governing when and how many patches to detach in each iteration? Did you perform ablation studies on these choices? How do different schedules (e.g., detach early vs later) affect both speed and quality?
>
> ## **Response**
> Thanks to the reviewer’s suggestion and our own curiosity, we could not resist conducting an ablation study on alternative detachment schedules. To this end, we compared two variants, **PRI (earlier)** and **PRI (later)**, against our original **PRI (uniform)** scheduling policy.
>
> All experiments use 4,000 inversion iterations, 75% sparsity (i.e. $v=4$), and DeiT-Small as the teacher model. The schedules are defined as follows:
> - **PRI (earlier)**: detachment occurs at iterations **[500, 1000, 1500, 4000]**
> - **PRI (uniform)**: detachment occurs at iterations **[1000, 2000, 3000, 4000]**
> - **PRI (later)**: detachment occurs at iterations **[2500, 3000, 3500, 4000]**
>
> The results are summarised below:
> | **Setting**            | **PRI (earlier)** | **PRI (uniform)** | **PRI (later)** |
> | ------------------ | ------------- | ------------- | ----------- |
> | Throughput (its/s) | 22.27         | 16.87         | 13.68       |
> | DeiT-Small $\rightarrow$ Tiny      | 63.99         | 64.77         | 57.52       |
> | DeiT-Small $\rightarrow$ Small     | 93.44         | 93.23         | 93.18       |
>
> Through this ablation study, we observed the following facts:
> - **Early detachment** maximises throughput by rapidly reducing the active patch set, yet still achieves accuracy close to the uniform schedule. This indicates that class-agnostic signals provide strong transferability even when class-specific refinement begins later. Interestingly, this also suggests that we can even find a better sweet spot in the trade-off between efficiency and accuracy, potentially enlarging the performance gap over SMI even further.
> - **Late detachment** preserves more class-specific structure for longer, but it brings neither speed gains nor accuracy improvements. This indicates that delaying patch removal offers limited practical benefit.
> - Overall, the **uniform schedule** remains a balanced choice: it preserves enough class-agnostic and class-specific information while maintaining competitive efficiency.

---

### Official Review · Reviewer_XhHS · 2025-11-04

**Soundness:** 3
**Presentation:** 3
**Contribution:** 2
**Rating:** 6
**Confidence:** 4

**Summary:**

The paper challenges the core assumption of Sparse Model Inversion (SMI) by showing that even randomly selected patches can acquire transferable knowledge through continued inversion. This motivates Patch Rebirth Inversion (PRI), a progressive strategy that:
- Dynamically detaches high-importance patches for sparse synthetic image construction .
- Keeps remaining patches evolving to accumulate class-relevant knowledge (the Re-Birth effect).

**Strengths:**

- Challenging SMI dogma: Demonstrates that discarding patches early is harmful, and even randomly selected patches can acquire transferable knowledge with continued optimization.

- This Re-Birth effect is novel, well-motivated, and empirically substantiated — a genuine contribution to understanding inversion dynamics in ViTs.

**Weaknesses:**

(1) I'm not sure if it's fair to make such a cross-sectional comparison between PRI and Sparse Model Inversion (SMI).
PRI allows patches to continue evolving after initial selection, effectively granting it more optimization steps per patch than SMI, which discards low-importance patches permanently. This violates the core constraint of SMI: fixed sparsity throughout inversion.
baseline would require:
- SMI with re-selection or reinitialization of discarded patches (currently absent)
- Or PRI with hard sparsity constraints (fixed patch set)
Without such controls, PRI’s efficiency advantage is confounded by increased computational flexibility.

(2) PRI reports results at 50% (v=2), 75% (v=4), 86% (v=7) — but v is undefined in the table.

(3) The negligible gains on ImageNet-1K (Table 3 a: +0.07 to +0.87 at matched sparsity) — while dramatic on CIFAR-100 distillation (Table 3 b: +6–19 pts) — reveal a critical scaling limitation of PRI.

**Questions:**

- How many total gradient steps are allocated to each patch across PRI vs. SMI?  For PRI: Do evolving patches receive additional updates beyond the initial fixed budget of SMI? If so, report per-patch update counts and total compute (FLOPs) for both methods.
- At convergence, what is the final active patch ratio in PRI? Is it strictly comparable to SMI’s fixed sparsity (e.g., 50%)?
- Compare PRI against an SMI variant with periodic re-ranking: every k steps, re-evaluate patch importance and reinstate top N (same budget as PRI’s dynamic set).  Does this closed-loop SMI close the gap in accuracy and speed?
- Show learning curves per patch type (detached early, late, or never) in PRI. Do randomly initialized “reborn” patches truly catch up in class relevance, or do gains come from over-optimizing a smaller elite set?

---

> ### Author Response · Authors · 2025-11-15
> **Response to (W1) and (Q3): Fairness of the comparison with SMI and SMI with periodic re-ranking**
>
> ## **Reviewer's concern**
>
> >I'm not sure if it's fair to make such a cross-sectional comparison between PRI and Sparse Model Inversion (SMI). PRI allows patches to continue evolving after initial selection, effectively granting it more optimization steps per patch than SMI, which discards low-importance patches permanently. This violates the core constraint of SMI: fixed sparsity throughout inversion.
> baseline would require:
> > - SMI with re-selection or reinitialization of discarded patches (currently absent)
> > - Or PRI with hard sparsity constraints (fixed patch set) Without such controls, PRI’s efficiency advantage is confounded by increased computational flexibility.
>
> > Compare PRI against an SMI variant with periodic re-ranking: every k steps, re-evaluate patch importance and reinstate top N (same budget as PRI’s dynamic set). Does this closed-loop SMI close the gap in accuracy and speed?
>
>
> ## **Response**
>
> We appreciate the reviewer’s concern regarding a potentially unfair comparison between PRI and SMI. We address this point in two parts.
>
> **(1) PRI does not receive more global optimisation steps than SMI.**
>
> In all experiments, PRI, SMI, and DMI share the same inversion schedule (number of iterations, optimiser, and learning-rate schedule). The "continued evolution" of patches in PRI is not a free extra budget, but rather it is fully counted in the inversion time and FLOPs reported in our efficiency tables (e.g., Table 2). In other words, PRI does not secretly receive more global optimisation steps than SMI, and the efficiency gains we report are measured after this budget is taken into account.
>
>
> **(2) SMI with periodic re-ranking is not part of the original SMI algorithm.**
>
> We agree that periodic re-ranking could in principle compensate for SMI’s fixed sparsity. However, this mechanism is not part of the original SMI design, which relies on: patch pruning without any re-selection or re-initialization. Adding periodic re-ranking fundamentally changes this assumption and effectively creates a new variant, rather than an adjustment of the existing baseline.
>
> **(3) Experimental result: SMI+ReRank does not close the gap.**
>
> Nevertheless, to fully address the reviewer’s suggestion, we implemented this variant of SMI, which we refer to as SMI+ReRank. In this variant, patch importance is re-evaluated every 1,000 iterations (as in PRI), and the top-N patches are re-selected under the same 75% sparsity constraint.
>
> | Method        | DeiT-Small $\rightarrow$ Tiny, Acc (%) | DeiT-Small $\rightarrow$ Small, Acc (%) |
> |---------------|---------------|----------------|
> | SMI           | 54.87          | 93.16          |
> | *SMI+ReRank*    | *55.97*          | *93.11*           |
> | PRI           | **64.77**          | **93.23**           |
>
> The results show that SMI+ReRank performs similarly to the original SMI version. PRI still outperforms both methods.

---

> ### Author Response · Authors · 2025-11-15
> **Response to (W2): v is undefined in the table.**
>
> ## **Reviewer's concern**
> > PRI reports results at 50% (v=2), 75% (v=4), 86% (v=7) — but v is undefined in the table.
>
> ## **Response**
> The variable $v$ is defined in Section 4 along with its role in determining the progressive sparsity schedule. However, we agree that the table caption should also restate this notation for clarity. We will revise the caption accordingly to make the definition of $v$ explicit.

---

> ### Author Response · Authors · 2025-11-15
> **Response to (W3): Differences in gains on ImageNet-1K vs. CIFAR-100**
>
> ## **Reviewer's concern**
> > The negligible gains on ImageNet-1K (Table 3 a: +0.07 to +0.87 at matched sparsity) — while dramatic on CIFAR-100 distillation (Table 3 b: +6–19 pts) — reveal a critical scaling limitation of PRI.
>
> ## **Response**
> We would like to clarify that the contrast in performance gains between Table 3(a) (ImageNet-1K quantization) and Table 3(b) (CIFAR-100 distillation) is not a scaling limitation of PRI, but instead reflects the fundamentally different difficulty levels and error margins of the two tasks.
>
> **(1) Quantization has an intrinsically small performance gap.**
>
> For ImageNet-1K quantization, the baseline 8-bit QAT already has strong self-recovery capability:
> - Full-precision model: 81.7%
> - 8-bit quantized model with DMI: 80.73%
>
> Because quantization causes only a minor degradation, the possible room for improvement is inherently limited. In such settings, even small gains like +0.07–0.87 represent meaningful improvements over a strong and stable baseline.
>
> **(2) Knowledge distillation is a substantially more difficult and lossy task.**
>
> In contrast, the DeiT-Base $\rightarrow$ DeiT-Tiny distillation setting creates a much larger information gap:
> - Teacher accuracy: 80.6%
> - Student with DMI: 54.9%
>
> This 26-point drop indicates that distillation suffers from severe information loss, meaning there is far more room for improvement. In this harsher scenario, PRI’s larger gains (+6 to +19 points) reflect the underlying task difficulty rather than any limitation in scalability.
>
> Therefore, the observed difference in improvements should be interpreted in the context of the intrinsic difficulty and error margin of each task, not as evidence of a scaling limitation of PRI.

---

> ### Author Response · Authors · 2025-11-15
> **Response to (Q1): Gradient steps and per-patch compute in PRI vs. SMI**
>
> ## **Reviewer's concern**
> > How many total gradient steps are allocated to each patch across PRI vs. SMI? For PRI: Do evolving patches receive additional updates beyond the initial fixed budget of SMI? If so, report per-patch update counts and total compute (FLOPs) for both methods.
>
>
> ## **Response**
> We would like to clarify that PRI does not allocate any additional optimization steps beyond the inversion budget used by SMI. As shown in Table 2, the total FLOPs and throughput of PRI are in fact lower than those of SMI, indicating that PRI operates under a stricter compute budget, not a looser one.
>
> A potential source of confusion is that PRI allows the active patch set to evolve during inversion. However, this does not increase the number of patches updated per iteration. In fact, the opposite is true.
> For example, at 75% sparsity:
> - **SMI** synthesizes **128 sparse images from 128 inversion images**, because each inversion image contains exactly one active sparse sample.
> - **PRI**, under the same sparsity, synthesizes the same **128 sparse images from only 32 inversion images**, because each inversion image produces four sparse samples (due to $v=4$).
>
> Thus, while PRI updates patches across multiple detachment stages, it updates fewer patches per iteration compared to SMI.
>
>
> To be precise, under 4,000 inversion iterations:
> - **SMI (76% sparsity)**: average per-patch update count $\approx$ 1055
> - **PRI (75% sparsity)**: average per-patch update count $\approx$ 2500
>
> Although PRI assigns more updates per patch, this does not imply higher computational cost. This is because SMI must synthesise a full 128 inversion images to produce 128 sparse samples, while PRI produces the same set of samples from only 32 inversion images, when $v=4$.
>
> Consequently, even with a larger per-patch update count, the overall FLOPs of PRI remain lower than SMI, consistent with Table 2 (also summarised below).
>
> | Sparsity | Method | FLOPs (T) | Δ vs. SMI |
> |----------|---------|-----------|-----------|
> | **50%**  | SMI     | 6.74      | —         |
> |          | PRI (v=2) | 5.02    | -25.5%    |
> | **~75%** | SMI (76%) | 3.45    | —         |
> |          | PRI (v=4, 75%) | 2.09 | -39.4% |
> | **86%**  | SMI     | 2.13      | —         |
> |          | PRI (v=7) | 1.07    | -49.8%    |
>
> This confirms that PRI does not rely on extra computation or additional optimization steps beyond SMI’s inversion budget.

---

> ### Author Response · Authors · 2025-11-15
> **Response to (Q2): Final active patch ratio in PRI**
>
> ## **Reviewer's concern**
> > At convergence, what is the final active patch ratio in PRI? Is it strictly comparable to SMI’s fixed sparsity (e.g., 50%)?
>
> ## **Response**
>
> Yes, for 50% sparsity, the final active patch ratio in PRI with $v=2$ is exactly comparable to SMI's fixed 50% sparsity.
>
> For higher sparsity levels (e.g., $v=4$), there is a small mismatch between the sparsity defined in SMI and the sparsity produced by PRI’s detachment schedule. As noted in Section 5, SMI’s predefined sparsity for this setting is 76%, whereas PRI naturally produces 75% sparsity under the same $v=4$ configuration.
>
> To ensure a fair comparison, we intentionally set $v=4$ so that PRI aligns as closely as possible with the original SMI setup, even though this results in a slightly stricter (more sparse) and therefore more challenging setting for PRI.
>
> This confirms that the two methods are directly comparable, and that PRI is evaluated under a marginally more difficult sparsity ratio than SMI.

---

> ### Author Response · Authors · 2025-11-15
> **Response to (Q4): Learning curves of different patch types in PRI**
>
> ## **Reviewer's concern**
> > Show learning curves per patch type (detached early, late, or never) in PRI. Do randomly initialized “reborn” patches truly catch up in class relevance, or do gains come from over-optimizing a smaller elite set?
>
> ## **Response**
>
> In the revised version of the paper, we include learning curves for different patch types in PRI (please refer to Figure A3 in the Appendix).
>
> As shown in Figure A3(b), models trained solely on the $t_4$ samples, which were considered less important at early stages, display a **fast and stable convergence** pattern. This demonstrates that these "reborn" patches indeed acquire class-relevant structure during inversion, rather than remaining weak or noisy.
>
> Moreover, we find that student models trained on individual groups $t_1$ through $t_4$ ultimately reach **similar final accuracies**. This indicates that PRI’s improvement does not come from over-optimizing a small elite subset of patches. Instead, each detachment group accumulates sufficient transferable information, and PRI benefits from combining these complementary patch groups rather than depending on any single one.

---

### Author Response · Authors · 2025-11-15

We would like to thank all four reviewers for reading our work so carefully. We addressed every question with detailed explanations and new experiments, and we added the necessary analyses to the Appendix to make our reasoning clear. We also appreciate the overall positive reception reflected in the reviews, which was very encouraging for us. We hope that our clarifications and additional results help all reviewers feel more confident about the contribution of this work.

---

### Author Response · Authors · 2025-11-27
**Follow-Up on Our Rebuttal Responses**

Dear Reviewers,

We sincerely apologize for the additional message. We fully understand that the reviewers are extremely busy, but we wished to gently follow up as we have carefully addressed all major points raised in the reviews. To summarise, we added:
- (1) fairness checks including the SMI+ReRank comparison
- (2) detailed analyses of compute, sparsity, and per-patch updates
- (3) learning-curve and feature-evolution results supporting the Re-Birth effect
- (4) ablations on detachment schedules and importance metrics.

If you might have a brief moment to look over our responses whenever it is convenient for you, we would be truly grateful. Thank you very much for your time and consideration.

Kind Regards,
Authors

---

### Author Response · Authors · 2025-11-29
**Rebuttal Summary for the AC**

Dear Area Chair,

Thank you very much for taking the time to evaluate our submission. Following the Program Chairs’ guidance, we provide a concise summary of (1) the strengths consistently highlighted across the four reviewers, and (2) the key clarifications and additional evidence we provided during rebuttal. We hope this summary helps streamline your assessment.

# 1. Strengths Commonly Recognised by Reviewers

- **Novel insight challenging SMI’s core assumption**

Reviewers (XhHS, UtL8, pfYh) emphasised the novelty of the Re-Birth effect, showing that patches deemed unimportant early can later acquire transferable knowledge, an insight not explored in prior sparse inversion work.

- **Simple, well-motivated, and intuitive method**

Reviewers (UtL8, UkW6) found PRI’s progressive detachment and continued evolution mechanism intuitive, novel, and appropriate for patch-based ViTs.

- **Strong efficiency and accuracy improvements**

Reviewers (UtL8, pfYh, UkW6) highlighted that PRI delivers: (1) higher throughput than both SMI and DMI, (2) significantly lower FLOPs, and (3) consistent accuracy gains across both distillation and quantisation.

- **Clear writing and clear supportive evidence**

Reviewer pfYh found the paper well-organised and easy to follow, while Reviewer UkW6 highlighted the combination of thorough theoretical analysis and empirical validation.

# 2. Key Clarifications and Additional Evidence Provided in the Rebuttal

- **Fairness & Compute Budget Clarification (XhHS)**
  * PRI uses *no additional global iterations* beyond SMI.
  * We provided explicit *per-patch update counts* and *FLOP comparisons* showing PRI is in fact more efficient.
  * We implemented reviewer-suggested **SMI+ReRank** variant; its performance remains *substantially below* PRI, confirming the fairness of the comparison.

- **Validation of the Re-Birth Effect (XhHS, UtL8)**

  * We have added or re-assured concrete evidence including:
    * learning curves of early/late/never-detached patch groups
    * t-SNE feature evolution analyses
    * transfer experiments per detachment stage, confirming that “reborn” patches genuinely acquire class-specific structure.

- **Robustness to Importance Metrics & Schedules (UtL8, UkW6)**
  * Gradient-based importance metric $\rightarrow$ performance nearly identical.
  * Ablations on early / uniform / late detachment schedules: uniform remains strongest and robust.


- **Scaling behaviour & dataset differences (XhHS, pfYh)**
  * ImageNet QAT has inherently small headroom (~1% drop), so gains must be small.
  * CIFAR distillation has a large teacher–student gap (~26 points), making PRI–SMI differences more visible.
  * Extended inversion (4k→8k→12k) shows SMI never approaches PRI, indicating intrinsic information loss from early patch removal.

- **Other minor clarifications (UtL8, UkW6)**
  * Complexity assumptions (SMI idealisation) clarified.
  * Generalisation beyond ViTs discussed.
  * Typos corrected.


Finally, we would like to emphasise that no reviewer identified technical flaws, and the rebuttal provided extensive new analyses that address all raised questions substantively.

We sincerely appreciate your time and hope this summary is helpful for your meta-review.

Kind regards,

The Authors

---

### Meta-Review · Area_Chair_YwDr · 2026-01-02

**Summary:**

This paper proposes Patch Rebirth Inversion (PRI), a novel model inversion framework for Vision Transformers that challenges the core assumption of prior sparse inversion methods. Instead of permanently discarding low-importance patches, PRI progressively detaches high-importance patches while allowing the remaining patches to continue evolving, enabling them to gradually accumulate transferable knowledge (the Re-Birth effect).

The paper received four reviews with mixed but overall positive scores, including three marginal accepts, and one borderline below threshold. The reviewers raised concerns regarding fairness of comparison with SMI, robustness of the patch-importance criterion, and generality beyond ViTs. After carefully reading the paper and the rebuttal, I believe the fairness concern is not fully resolved, as PRI can be viewed as repeatedly applying SMI over different subsets of patches, which complicates a clean methodological comparison. In addition, the technical contribution remains limited, as key components such as the importance metric and related design choices largely follow existing prior work rather than introducing fundamentally new algorithmic elements.

**Reviewer Concerns:**

Reviewer XhHS questioned whether the comparison between PRI and SMI is fair, noting that PRI allows patches to continue evolving while SMI enforces fixed sparsity. The rebuttal clarified that PRI does not use additional global optimization steps, provided detailed FLOP and per-patch update analyses, and implemented an SMI+ReRank baseline, which still underperformed PRI. However, despite having the same number of global optimization steps, PRI allocates more optimization steps to different subsets of patches over time, which makes the comparison with SMI not fully fair and does not clearly demonstrate an inherent algorithmic advantage of PRI.

Reviewer UtL8 raised questions about the robustness of the patch-importance metric, and the grounding of the Re-Birth effect. The authors provided additional ablations with gradient-based importance metrics, and extensive feature-evolution analyses. These concerns were well addressed.

Reviewer pfYh asked whether different sparsification methods would converge to similar performance given sufficient training time and why PRI shows larger gains in some settings. The rebuttal included extended inversion and training experiments demonstrating that SMI does not close the gap even with significantly longer training. However, these results do not clearly isolate the algorithmic advantage of PRI, as the observed gains could still be attributed to differences in optimization dynamics.

Reviewer UkW6 raised minor concerns regarding idealized complexity assumptions for SMI, choice of importance criteria, and intuition behind patch rebirth. The rebuttal clarified these assumptions, added robustness experiments, and provided intuitive explanations and additional analysis. These concerns were largely resolved.

**Reviewer Scores:**

The paper received three marginal accepts and one borderline below threshold.

Reviewers UtL8, and UkW6 would likely maintain their positive scores after the additional analyses and experiments.

Reviewer XhHS may lower their score, as concerns regarding the fairness of the comparison and the algorithmic contribution of PRI are not fully addressed in the rebuttal.

Reviewer pfYh is likely to maintain a borderline reject score, as the source of the observed performance gains is not clearly explained from an algorithmic perspective.

---

### Decision · Program_Chairs · 2026-01-26

Reject